# Acquired Bacterial Resistance to Antibiotics and Resistance Genes: From Past to Future

**DOI:** 10.3390/antibiotics14030222

**Published:** 2025-02-21

**Authors:** Michela Galgano, Francesco Pellegrini, Elisabetta Catalano, Loredana Capozzi, Laura Del Sambro, Alessio Sposato, Maria Stella Lucente, Violetta Iris Vasinioti, Cristiana Catella, Amienwanlen Eugene Odigie, Maria Tempesta, Annamaria Pratelli, Paolo Capozza

**Affiliations:** 1Istituto Zooprofilattico Sperimentale della Puglia e della Basilicata, 71121 Foggia, Italy; michela.galgano@izspb.it (M.G.); elisabetta.catalano@izsbp.it (E.C.); loredana.capozzi@izspb.it (L.C.); laura.delsambro@izspb.it (L.D.S.); alessio.sposato@izspb.it (A.S.); 2Department of Veterinary Medicine, Università Aldo Moro di Bari, 70010 Valenzano, Italy; francesco.pellegrini@uniba.it (F.P.); mariastella.lucente@uniba.it (M.S.L.); violetta.vasinioti@uniba.it (V.I.V.); cristiana.catella@uniba.it (C.C.); amienwanlen.odigie@uniba.it (A.E.O.); maria.tempesta@uniba.it (M.T.); annamaria.pratelli@uniba.it (A.P.); 3Department of Public Health, Experimental and Forensic Medicine, University of Pavia, Via Carlo Forlanini 2, 27100 Pavia, Italy

**Keywords:** bacteria, antibiotic resistance, antibiotic resistance genes, health emergency

## Abstract

The discovery, commercialization, and regular administration of antimicrobial agents have revolutionized the therapeutic paradigm, making it possible to treat previously untreatable and fatal infections. However, the excessive use of antibiotics has led to develop resistance soon after their use in clinical practice, to the point of becoming a global emergency. The mechanisms of bacterial resistance to antibiotics are manifold, including mechanisms of destruction or inactivation, target site modification, or active efflux, and represent the main examples of evolutionary adaptation for the survival of bacterial species. The acquirement of new resistance mechanisms is a consequence of the great genetic plasticity of bacteria, which triggers specific responses that result in mutational adaptation, acquisition of genetic material, or alteration of gene expression, virtually producing resistance to all currently available antibiotics. Understanding resistance processes is critical to the development of new antimicrobial agents to counteract drug-resistant microorganisms. In this review, both the mechanisms of action of antibiotic resistance (AMR) and the antibiotic resistance genes (ARGs) mainly found in clinical and environmental bacteria will be reviewed. Furthermore, the evolutionary background of multidrug-resistant bacteria will be examined, and some promising elements to control or reduce the emergence and spread of AMR will be proposed.

## 1. Introduction

The use of chemical molecules to counteract diseases, especially infectious ones, dates to Ancient Egypt, where home-made remedies were identified for the treatment of open wounds, i.e., the application of molds, the ingestion of radishes, leeks, garlic, and onions, today considered antibacterials [1,2]. At the end of the 19th century, the discovery of bacterial antagonism, understood as the ability of one micro-organism to interfere with the growth of another micro-organism, such as the ‘inhibition of Bacillus anthracis’ by micro-organisms in the air or the antagonistic action of *Penicillium* spp. on bacterial growth in infected wounds, was the prelude to laying the foundations for the development of antibiotics [3,4]. The concept of microbial antagonism, however, was only understood thanks to the works of researchers, was subsequently supported by the experiments, and was finally accepted by the scientific community with the name of “antibiosis” [1,5,6,7].

The beginning of a new era of innovation in the use of drugs for human and animal health was marked by Alexander Fleming’s studies, which, by co-inoculating a lethal dose of *Salmonella typhi* with *Penicillium glaucum*, managed to counteract the development of typhoid, demonstrating the clinical importance of penicillin in preventing the development of the disease [2,8] (Figure 1).

## 2. Resistance or Persistence?

Antimicrobial resistance is one of the major obstacles to the successful treatment of microbial infections, although we are not entirely sure that it is always “antibiotic resistance”. We must therefore make a distinction between resistance and persistence. Failure of antimicrobial therapy may be caused by “truly” resistant bacteria to the antimicrobial agent or by the presence of persistent cells, i.e., cells in the stationary growth phase (not actively dividing) on which most antimicrobial agents have no effect. The presence of these cells (in stationary phase) in a bacterial culture is estimated at a rate of about 1% [9,10]. Only when replication/division resume will the persistent cells develop sensitivity to the antimicrobial agent, since antibacterials act only on metabolically active and replicating cells [11]. The concept of antibiotic persistence was first developed in the 1940s by Hobby and Bigger, who had noted the survival of small fractions of bacteria after exposure to normally bactericidal concentrations of penicillin, which were shown to be sensitive to the same concentrations of antibiotic after being re-cultured on a new culture substrate. This prompted the hypothesis that penicillin only kills growing cells [12,13]; the surviving fraction could have escaped death by entering some type of dormant or non-growing state. This fraction of surviving cells referred to as ‘persisters’ (metabolically quiescent cells of a genetically susceptible bacterial population) is to date, according to several studies, potentially related to antibiotic resistance, as persisters provide a viable cell reservoir from which resistant mutants can emerge via horizontal gene transfer or de novo chromosomal mutations [14].

Another important concept regarding the persistence of certain bacterial strains is the classification of antibiotics according to their mechanism of action as ‘bactericidal’ and ‘bacteriostatic’. A ‘bacteriostatic agent’ only halts the growth of bacteria without necessarily killing the microorganisms. This would inhibit the multiplication and spread of bacteria, keeping the population constant, working through inhibition of bacterial protein synthesis, nucleic acid synthesis, or other crucial metabolic pathways. In contrast, a bactericidal agent actively kills by disrupting vital cellular processes or damaging bacterial structures and bacteria beyond repair and reduces the bacterial population [14].

Indeed, it has recently been shown that static antibiotics can positively influence the quiescent state in bacteria, resulting in an increase in the persistent fraction in a bacterial culture, and favor the development and spread of antibiotic resistance [15]. In Table 1, the antibiotics groups divided by mechanism of action were listed.

## 3. Antimicrobial Resistance Mechanisms

The study of the mechanisms of antibiotic resistance of bacteria belonging to the same group or species has revealed non-uniform resistance capacities and levels, which can be considerable even within the same bacterial groups or species [10]. This variability is the consequence of several factors, including selective pressure due to the frequency of exposure to antimicrobials and dosages [10]. The emergence of antimicrobial resistance has greatly affected the impact of infectious diseases, in terms of the number of infections, as well as increased healthcare costs. The report, published by Mohsen et al. (2024) [16], found that between 1990 and 2021, more than one million people die each year from drug-resistant infections. Many of the deadliest infections in this time frame were caused by *E. coli* and *S. aureus*. In fact, according to the 2022 Global Antimicrobial Resistance and Use Surveillance System (GLASS) report, 35% of *Staphylococcus aureus* is resistant to methicillin, while 42% of *E. coli* is resistant to third-generation cephalosporins and shows reduced susceptibility to standard antibiotics, such as ampicillin, co-trimoxazole, and fluoroquinolones, thus making it increasingly difficult to effectively treat common infections [17]. In addition, according to the Centers for Disease Control and Prevention, more than two million people in the United States fall ill with antibiotic-resistant diseases each year, causing a minimum of 23,000 deaths [18]. Making the estimate of current deaths attributed to AMR even more worrying, a review by O’Neill et al. (2024) [19], in which it was estimated that antimicrobial resistance could cause 10 million deaths per year by 2050, is reported in the literature. At the economic level, the high cost of healthcare due to antimicrobial resistance in Europe, it has been estimated that antimicrobial resistance has been correlated with more than nine billion euros per year, and in the United States, it is estimated at 20 billion dollars for the health sector and about 35 billion dollars for the loss of productivity due to this crisis of antimicrobial resistance [20,21].

Although we have a wide range of choices, a documented antimicrobial resistance to all of these exists, resistance that arises shortly after approval for use. In fact, after the introduction of an antibiotic into clinical practice, the development of a resistance mechanism has been shown to be an inevitable consequence of bacterial evolution; hence, the World Health Organization (WHO) developed in 2015 a Global Action Plan on Antimicrobial Resistance [22].

Bacteria’s goals are to replicate, survive, and spread as quickly as possible, adapting, evolving, and counteracting drugs through genetic changes that ensure their continued existence [23]. This natural process by which bacteria develop drug resistance has been exacerbated by the overuse and abuse of antibiotics, inaccurate diagnoses, improper prescription, self-medication, inadequate healthcare environments, poor hygiene, and their widespread use also in agriculture [24,25]. However, there is strong evidence that antibiotic resistance emerged and evolved long before the use of synthetic antibiotics by humans in the 1940s [25,26] and was ubiquitous in intact ecosystems with limited or no anthropogenic disturbance/interference [27,28,29].

Indeed, the soil biota are hypothesized to exert selective pressure on bacterial antibiotics production, probably due to the presence of natural antibacterial molecules, which even at low concentrations can promote the emergence, acquisition, and spread of environmental ARGs among bacteria. These “environmental resistomes”, perhaps attributed to telluric microorganisms producing antibiotics or their derivatives presumably to compete with other organisms living in the same antibiotic environment, have exerted selective pressure on the soil microbiome [30]. These biotic interactions between bacteria and other microorganisms can directly influence bacterial community composition, promoting the abundance of antibiotic resistance genes (ARGs) [31]. These bacteria represent a prolific source for the acquisition and dissemination of antibiotic resistance genes in clinically relevant bacteria and thus pose a global threat to human health [32].

The elements that contribute to the multifaceted etiology of antibiotic resistance can be classified as natural or acquired (chromosomal or plasmid). Natural resistance can be intrinsic or induced (after exposure to an antibiotic). Intrinsic resistance can be defined as a trait that is universally shared within a bacterial species and independent of previous antibiotics exposure (i.e., resistance to vancomycin and ampicillin in *Escherichia coli*, resistance to first- and second-generation cephalosporins in *Pseudomonas aeruginosa* [33]). Induced resistance is the phenotypic expression of pre-existing resistance genes that alter membrane permeability and efflux pump activity [34,35]. However, the resistance mechanisms are different between Gram-positive and Gram-negative bacteria due to their structural differences [34]. Gram-positive bacteria, characterized by an outer layer of coarsely meshed peptidoglycan adorned with teichoic acid polymers and covalently bound proteins, facilitate the passage of small molecules of 30–57 kDa [36] and are relatively susceptible to various antibiotics [37]. Gram-positive resistance is generally related to mechanisms involving destruction or inactivation, changes in target site, or active efflux [38], which less commonly use drug uptake restriction, due to the lack of an outer LPS membrane and the lack of capacity for some types of drug efflux mechanisms [39]. Conversely, Gram-negative bacteria have an “outer membrane” (OM) whose lipid molecules have several fatty acids chains that contribute to reducing membrane fluidity and thus to reducing the permeability threshold, making the bacteria intrinsically insensitive to many different antibacterial agents [40]. However, despite the relative impermeability of the OM, Gram-negative bacteria possess other mechanisms that allow for the passage of nutrients. The OM is in fact dotted with a variety of proteins belonging to the class of porins, which can be separated into two subgroups: porins permeable to general diffusion, based on the size of the molecules that pass through them, and porins that are more solute-specific, which have binding sites within them that, depending on the charge and size of the solute, promote the passage of selected substrates necessary for bacterial growth [41]. In addition, these channels also appear to be involved in the phenomenon of antibiotic resistance by limiting the influx of many antibiotics, such as β-lactams, fluroquinolones, and tetracyclines, thus contributing to the intrinsic antibiotic resistance of many microorganisms [42]. In addition to intrinsic resistance mediated by OM membrane pumps and efflux (active efflux), the identification of a considerable number of additional genes and genetic loci contributes to the multidrug-resistant (MDR) phenotype exhibited by bacteria [34]. This phenotype, called the “intrinsic antibiotic resistome” and its identification as an ancient natural phenotype present in all bacterial species long before the use of antibiotics, has shaken the long-held belief that antibiotic resistance originated from human activity and clinical experimentation with antibiotics [43,44]. However, it remains a controversial topic that requires further studies to demonstrate its origin and its ability to make bacteria hypersensitive to antibiotics, suggesting that through the inhibition of these targets, the activity of several antibacterial molecules could be restored, improving the therapeutic response against pathogenic bacteria [34]. Acquired resistance can develop either through mutations or genetic recombinations or by acquisition of genes that can be performed by both vertical gene transfer (VGT) and horizontal gene transfer (HGT). DNA mutations (substitutions, deletions, etc.) are usually caused by stressors (starvation, UV radiation, chemicals, etc.) and can be transferred to bacterial offspring. The average mutation rate in bacteria is estimated 1 per 10^6^–10^9^ cell divisions; however, most of them are deleterious to the cell. In fact, resistance-associated mutational changes are, in many cases, burdensome for cellular homeostasis (i.e., cause a decrease in fitness) and therefore are maintained only, when necessary, in the presence of the antibiotic [45,46]. For example, following the acquisition of methicillin resistance by *Staphylococcus aureus*, a significant reduction in the growth rate of the bacterium is observed [33]. However, this weakness is compensated by the selection of a resistant mutant in the bacterial community that becomes predominant following the elimination of the entire susceptible population.

Among the most important factors in bacterial evolution for the development of antimicrobial resistance, the indiscriminate use of antibiotics and/or the use of low concentrations of antimicrobials (sub-inhibitors) is the one with the most important implications for the selection of hypermutable strains (increased mutation rate) and increase the incidence of acquired resistance by bacteria to antimicrobial agents [47]. Once the mutation is acquired, VGT and HGT become the most noteworthy mechanisms for the spread of amicrobial resistance [48].

VGT is a mechanism in which the drug resistance gene is transmitted during the process of bacterial division from parent to offspring over generations. Meanwhile, HGT allows for the exchange of genes between different species, acquiring an important role in the evolution and spread of bacterial multidrug resistance [48]. This type of genetic recombination occurs through the processes of transformation, direct transfer of extracellular DNA, transduction, phage-mediated extracellular DNA transfer, conjugation, and bacterial sex-pilot transfer [36]. Transformation is perhaps the simplest type of HGT, but only a few clinically relevant bacterial species are able to “naturally” incorporate exogenous DNA. However, it remains a critically important phenomenon given its contribution to the spread of antibiotic resistance. Conjugation is a very efficient method of gene transfer and is likely to occur in the gastrointestinal tract, given the assortment of bacterial groups, during antibiotic therapy at high rates and with increased efficiency. Typically, conjugation uses mobile genetic elements (MGEs), such as plasmids and transposon elements, to transmit valuable genetic information for bacterial evolution [49,50]. Plasmids, small strands of supercoiled double-helix DNA, have been found in almost all bacterial genera and can transfer MGE and genes encoding antibiotic resistance to new hosts via conjugation [51,52]. The ability of a plasmid to transfer genetic material from one species to another can be more or less specific. In fact, there are some plasmids capable of conjugating with a range of hosts limited to one genus or species, while others have a broader host range of and can transfer from one species to another less selectively. In some cases, plasmids transferred to a particular host are unable to replicate efficiently. In these circumstances, the genetic material can be maintained only if it is contained in a transposon; this genetic element can translocate into the bacterial chromosome and be maintained, while the plasmid is lost. This additional ability to transfer genetic material makes the presence of the plasmid in the specific host unnecessary to contribute to the spread of resistance [53].

Transposons, also referred to as transposable elements or ‘jumping genes’, are a heterogeneous set of small genomic elements that, in the presence of specific enzymes called transposases, move from one point on a chromosome (donor site) to another on the same or another chromosome (target site). Transposition of the transposon can occur on the same DNA molecule or on a different one. Transposons can be found as part of the nucleoid of a bacterium (conjugative transposons) or in plasmids and are typically one to twelve genes long [54]. A transposon can contain several genes and are always flanked at both ends by insertion sequences encoding the enzyme transposase, which catalyzes the cutting and resealing of DNA during transposition. Therefore, such transposons can cut from a bacterial nucleoid or plasmid and insert into another nucleoid or plasmid, contributing to the transmission of antibiotic resistance in a wider range of hosts than plasmids [55]. Bacterial transposons belong to two classes: compound transposons (class I) and complex transposons (class II). The class I transposon, containing a copy of a terminal insertion sequence (IS), characterized by repeated and inverted sequences at both ends, also provides two transposases, each of which has the potential to direct site-specific recombination, facilitating the possibility of being mobilized and thus contributing to the spread of antibiotic resistance genes. The class II transposon has at its ends two identical sequences of 30 to 40 nucleotide pairs each, arranged in opposite orientation, which are unable to transfer conjugally to other bacteria, and must be contained within a conjugative element to be disseminated [56,57].

Finally, another efficient mechanism for the acquisition of antimicrobial resistance genes is represented by integrons, genetic elements of a site-specific recombination system capable of adding new genes into bacterial chromosomes, together with the mechanisms necessary to ensure their expression [50]. Unlike transposons, integrons do not have repeated sequences and do not include genes coding for proteins responsible for their movement towards other bacteria, mostly mediated by plasmids or transposons [58]. They carry an integron gene, which encodes the site-specific integrase used for categorizing integrons into “class” [59]. Four classes are now recognized: class 1 (intI1), class 2 (intI2), class 3 (intI3), and class 4 (intI4) [60,61]. The brief description differentiating the classes of integrons is given below. Classes 1, 2, and 3 are commonly identified in clinical settings, while class 4 is associated with the SXT element found in *Vibrio cholerae* (Flui [62]. The class 1 integron is not automotive, while other mobile genetic elements, such as conjugative plasmids and associated transposons, are capable of acting as vehicles for intra- and inter-species transmission of genetic material. It is the most ubiquitous and most commonly reported among clinical Gram-negative bacteria (*Escherichia* spp., *Klebsiella* spp., *Pseudomonas* spp., *Salmonella* spp., *Staphylococcus* spp., *Enterococcus* spp., and *Vibrio* spp.). Similar to the class 1 integron organization, the class 2 integron is commonly associated with the transposon family that mediates integron mobility. It has been commonly reported in some species of Gram-negative organisms, such as Acinetobacter spp., Salmonella spp., and Psuedomonas spp., but with a lower incidence and prevalence than the class 1 integron. The structure of class 3 integrons is comparable to that of class 2 integrons. Its identification has been limited to a few microorganisms, including *Acinetobacter* spp., *Citrobacter freundii*, *Escherichia coli*, *Klebsiella pneumoniae*, *Pseudomonas aeruginosa*, *Pseudomonas putida*, *Salmonella* spp., and *Serratia marcescens*, with low detection rates. Class 4 integrons are more complex structures with a large number of cassettes, with around 130 different resistance genes identified, reflecting a wide phylogenetic diversity but not normally associated with antimicrobial resistance. Its detection has been limited within microorganisms, such as Vibrionaceae, Shewanella, Xanthomonas, Pseudomonad, and other proteobacteria.

## 4. Factors Affecting the Acquisition of Resistance Genes

The ability of mobile genetic elements containing antibiotic resistance genes to spread is modulated by several factors, such as (i) host-encoded specific factors, (ii) non-specific host factors, (iii) genetic-element-encoded factors, and (iv) environmental factors.

(i)Host-encoded specific factors: There are several systems with which bacteria protect themselves from exogenous DNA. The most common are restriction/editing systems and CRISPR-Cas systems (acronym for regularly interspaced short palindromic repeats/CRISPR-associated protein), immuno-adaptive defense mechanisms used by archaea and bacteria capable of identifying and degrading incoming foreign genetic material [63,64]. Both systems can reduce the spread of phage DNA, integrative conjugative elements (ICEs), and plasmids.(ii)Non-specific host factors: In this case, bacteria do not possess the species-specific target site for a given integrative conjugative element (ICE), or host replication systems prevent plasmid replication. In addition to endogenous systems, cell surface architecture may also hinder conjugation by reducing the productive functionality of mobile genetic element transfer. Furthermore, Gram-positive and Gram-negative bacteria produce a wide range of inhibitory substances and antimicrobial products to protect themselves from the constant assault of bacteriophages, the most common of which are the colicin bacteriocins produced by *E. coli* [65].(iii)Genetic element-encoded factors: To overcome bacterial defense systems, mobile genetic elements such as plasmids and ICEs can encode anti-restriction proteins that inactivate the host’s restriction system, allowing the MGE to enter the new host without being degraded. In addition, some genes encode anti-restriction proteins that mimic the structure of DNA, exhibiting DNA-like negative surface charge distributions, which are recognized and bound by the restriction enzyme [66].(iv)Environmental factors: The gene transfer for the spread of resistance is influenced by the presence of antibiotics in the environment and is favored in environments with relatively high density, such as the intestine and oral cavity, or in biofilm.

From the above, it can be deduced that to control the spread of resistance, it is necessary to know the biological mechanism of the different mobile genetic elements and the environmental ecosystem in which they develop and spread

## 5. Mechanism of Resistance

Bacteria have evolved sophisticated drug resistance mechanisms that likely developed over millions of years of evolution. These strategies include specific biochemical pathways, and, in general, antimicrobial resistance mechanisms are based on the limitation of drug absorption, drug inactivation, and drug efflux, typical examples of intrinsic resistance, while the acquired resistance mechanisms used can go beyond drug inactivation, drug efflux, and modification of the drug target [33]. Bacterial-resistance-inducing enzymes usually belong to large superfamilies, which include enzymes of different natures, and some of them have originated from enzymes that had other metabolic functions [55]. The genes encoding these enzymes are located on mobile genetic elements, some of which are conjugative plasmids and/or transposons, and are often associated with other resistance genes [67].

The most common mechanisms can be grouped into four broad categories based on the metabolic pathway involved: (i) structural modifications of the antimicrobial molecule (drug inactivation), (ii) preservation of the antibiotic target site (by reducing penetration or actively expelling the antimicrobial compound), (iii) alterations and/or bypass of the target sites, and (iv) global cellular adaptive process [34,68,69] (Figure 2).

(i)
*Structural modifications of the antimicrobial molecule*


The inactivation of a pharmacologically active substance represents one of the most effective strategies used by bacteria. This inactivation can occur mainly in two ways: (a) by adding specific chemical groups/functional groups to the compound or (b) by degrading the structure of the molecule, rendering it ineffective for its target (“Corpora non agunt nisi fixata”) [68,70].

(a)The production of enzymes capable of introducing chemical changes in the anti-microbial molecular structure is a mechanism of acquired resistance long known in both Gram-negative and Gram-positive bacteria. There are several modifying enzymes that catalyze the reactions through acetylation (AAC, Aminoglycoside N-acetyltransferase; CAT, Chloramphenicol acetyltransferase; VAT, virginiamycin O-acetyltransferase), phosphorylation (APH, Aminoglycoside phosphotransferase; CPT, Chloramphenicol O-phosphotransferase), and adenylation (ANT, Aminoglycosides adenylyltransferase, LIN Lincosamide adenylyltransferase). These enzymes belong to the transferase family, a large superfamily of enzymes that differ in terms of substrate specificity and mechanism of action, capable to covalently bind various chemical groups [67,71,72]. Regardless of the biochemical reaction, the resulting effect is often related to a steric disorder that decreases the drug’s avidity towards its target [73].

Enzymatic action aims to inactivate the active site of the drug, and one of the most relevant examples of resistance through drug modification is represented by aminoglyco-side modifying enzymes (AMEs) capable of altering the aminoglycoside molecule in a specific position, through covalent modifications of specific functional groups (hydroxyl or amino groups) of these antibiotics. To date, several hundred different AMEs have been described, and almost every enzyme is composed of several isoenzymes specific for a single substrate and modifying it in certain positions [62]. The rapid spread of *ame* genes determined by their location on mobile genetic elements, which favors their transmission and spread, makes this aminoglycoside resistance mechanism prominent throughout the world.

Three families of AMEs are distinguished: N-acetyltransferases (AAC), O-phosphotransferases (APH), and O-adenylyltransferases (ANT). The geographical distribution, the bacterial groups and species in which these enzymes are present and/or are able to spread, and the specific aminoglycosides that they modify are remarkably heterogeneous within the ecological community.

For example, the family of aminoglycoside 3′-phosphotransferases [APH(3′)], capable of altering the aminoglycoside molecules kanamycin and streptomycin but not gentamicin and tobramycin, is widely distributed in several bacterial species, including both Gram-positive and Gram-negative [74]. Although ecologically ubiquitous, a high presence of this gene has been described in Gram-negative bacteria in China (*Escherichia coli* and *Acinetobacter baumannii*), in Malaysia (*Enterococcus faecalis* and *Enterococcus faecium*), and in Egypt and Kentucky (*Klebsiella pneumoniae*), while in Gram-positive bacteria, the highest detection rate has been recorded in China and Africa from hospital isolates of *Staphylococcus aureus* [75,76]. Meanwhile, in the AME family, differences in the activity and distribution of aminoglycoside-mediating enzymes have been recorded. In fact, an example is given of the aminoglycoside O-nucleotidyltransferases (ANTs), which is capable of inactivating both gentamicin and tobramycin by catalyzing nucleotidylation modifications of the hydroxyl group in positions 4, 6, and 9, depending on the class they belong to, ANT (4′), ANT (6′), and ANT (9′), respectively. Furthermore, different locations of the genes encoding ANT (4′), ANT (6′), and ANT (9′) have been documented within bacterial communities, which, in Gram-positive microorganisms, are usually related to the bacterial MGEs, while, in Gram-negative bacteria, mainly *ant* (2″) and *ant* (3″) genes were found that were not associated with the MGEs but were located on gene cassettes belonging to class 1 integrons [74].

Another example of enzymatic alteration is the modification of the chloramphenicol molecule (Cm) by N-acetyltransferase (AAC which catalyzes the acetylation of acetyl-CoA at the 3-hydroxyl group of chloramphenicol or its synthetic analogs (thiamphenicol, azidamphenicol). This process prevents the binding of the active ingredient to bacterial ribosomes, thus compromising their efficacy [71]. The genes *aac* can be located on chromosomes [77] or more commonly can be located on plasmids as an integral part of transposons in association with other genes encoding antibiotic resistance [72].

In addition to the acetylation mechanism, chloramphenicol inactivation can be achieved by O-phosphorylation. This resistance mechanism has been described in a Cm-producing streptomyces, *S. venezuelae*, capable of modifying the same functional group by an alternative mechanism. Some studies hypothesize that this phosphorylation pathway is implicated in self-resistance as well, even in the horizontal transfer mediated by plasmids when introduced by transformation into the Cm-susceptible host [78].

Enzymes that modify macrolide, lincosamide, and streptogramin (MLS) antibiotics include phosphotransferases (MPH), glycosyltransferases, and acetyltransferases (macrolides, ketolides, lincosamides, and streptogramins). As with phosphodiesterase, phosphotransferases are enzymes capable of inactivating macrolides by interacting directly with the active site of the molecule, via the transfer of the γ-phosphate group of the nucleotide triphosphate to the 2′-hydroxyl group (2′-OH group) of the 14-, 15-, and 16-membered lactone macrocyclic rings, thereby modifying their biochemical structure by rendering them unable to interact with the ribosomal targets [71,79]. Seven different enzymes of this group have been described so far, divided into MPHA, which has the highest affinity for 14- and 15-membered ring macrolides, and MPHB, which preferentially catalyzes the phosphorylation of groups 2′-OH of the molecules 14 and 16 members [80]. The genes encoding MPHs are located on mobile genetic elements, commonly containing other genes that encode not only resistance to macrolides but also to other classes of antibiotics [81], and their expression may be inducible (*mph*A) or constitutive (*mph*B) [80].

Macrolide glycosyltransferases, on the other hand, inactivate macrolides by glycosylating the 2′-OH group of the macrolide ring, while Streptogramin acetyltransferases are able to activate streptogramin A only by acetylation of an unbound hydroxyl group [59]. The genes that encode these enzymes, sat genes, have been identified on plasmid MGEs in numerous Gram-positive pathogens, including staphylococci (e.g., *S. aureus* and *S. cohnii*) and enterococci (e.g., *E. faecium*), while, in Gram-negative bacteria, they are located on the chromosome, as in the case of *P. multocida*, *Yersinia enterocolitica*, and *S. putrefacients* [60]. In this category of modifying enzymes, molecules that inactivate fosfomycin have also been reported. To date, in bacteria that are resistant to fosfomycin, three types of metalloenzymes (FosA, FosB, or FosX) and two kinases (FomA and FomB) have been found [68]. FosA is a glutathione S-transferase which is encoded by the *fos*A-like gene, is located on plasmids or chromosomes, and is able to catalyze the addition of the glutathione molecule to the oxirane ring of Fosfomycin [82]. FosB is a thiol S-transferase, expressed by *fos*B-like genes, which facilitates the incorporation of a thiol group using bacillithiol, the α-anomeric glycoside of l-cysteinyl-d-glucosamine with l-malic acid, as the donor substrate [83]. FosX is an epoxide hydrolase that facilitates the hydration of fosfomycin, cleaving the oxirane ring and producing a diol product [84]. FomA transforms fosfomycin into fosfomycin monophosphate, whereas FomB generates fosfomycin diphosphate using the monophosphate form as a substrate [85].

For rifamycins, environmental bacteria have developed several mechanisms to inactivate rifamycins, including ADP-ribosyltransferases [86], glycosyltransferases [87], phosphotransferases [88], and monooxygenases [89]. These enzymes cause a change in the characteristic basket-shaped structure of the rifamycins, which is the key group involved in the binding of an antibiotic molecule to the β-subunit of RNA polymerase, resulting in a decrease in the affinity of rifampin to its binding site and its mechanism of action/efficacy.

The TetX monooxygenase and monooxygenase Rox, belonging to the redox family enzyme, give resistance to all tetracyclines. TetX catalyzes tetracycline monohydroxylation, leading to intramolecular cyclization and molecule decomposition, including broad-spectrum antibiotic tigecycline [71]. Rox is able to inactivate the active principles contained in rifamycins by oxidizing the naphthyl group in position 2, resulting in ring opening and linearization of the antibiotic molecule [90]. For antibiotics formulated in prodrug forms that are modified into active forms by lyases enzymes that catalyze the cleavage of various chemical bonds to form a new double bond or structural ring [91], the antimicrobial resistance is attributed to mutations in the genes encoding these enzymes, which prevent the prodrug from being converted into an active form. For example, for isoniazid, ethionamide, and prothionamide, which are activated by the NADPH-dependent FAD-containing monooxygenase encoded by the *eth*A gene, resistance is caused by mutations in the *eth*A and *eth*R genes, which result in conformational changes in the binding pocket, which reduces the affinity for the molecule which cannot exert its mechanism of action.

(b)The mechanism of destruction of antibiotic molecules involves the inactivation of the antibiotic’s active ingredient through their degradation. β-lactamases and macrolide esterases that destroy β-lactams and macrolides, respectively, are the most common enzymes that catalyze antibiotic hydrolysis. β-lactamases, first described in the early 1940s, belong to a superfamily of enzymes that currently has more than 2000 members [92]. They are responsible for the hydrolysis of the amide bond in the β-lactam ring, the common structural element of all β-lactam antibiotics (penicillins, cephalosporins, carbapenems, and monobactams) [93]. In an attempt to group this large number of enzymes, two main, not entirely overlapping, classification schemes have been proposed: (1) the Ambler classification, based on amino acid sequence identity and separating the β-lactamases into four groups, in which the enzymes of classes A (CTX-M, Cefotaximase; TEM, Temoniera of the patient in which it was originally found; SHV, variable sulfhydryl reagent; and KPC, *Klebsiella pneumoniae* carbapenemase), C (BLC, C β-lactamases, also known as AmpC or cephalosporinase), and D (OXA, Oxacillinase) are classified as serine hydrolases, while the enzymes of class B (NDM, New Delhi metallo-β-lactamase; VIM, metallo-β-lactamase encoded by the Verona integron; and IMP, imipenemase) are metalloenzymes; (2) the Bush-Jacoby classification, which divides β-lactamases into four categories (each with different subgroups) based on their biochemical function, mainly based on substrate specificity [94,95]. The high mutation rate of β-lactamases contributes to the rapid spread of resistant bacteria [94]. Also noteworthy and posing a global threat is the detection in hospital clinical samples of pathogenic bacteria carrying up to eight β-lactamase genes simultaneously, which makes them capable of hydrolyzing most penicillins and cephalosporins, adversely affecting clinical and therapeutic outcomes, with higher rates of morbidity and mortality, longer hospital stays and high healthcare costs [96]. In 2004, a strain of Klebsiella pneumonia was described, which was isolated from seven New York City hospitals and produced up to 10 different β-lactamases, including a FOX-like plasmid-mediated AmpC, in addition to the previously reported KPC, SHV ESBL, and IRT β-lactamases [97].

Class A β-lactamases include CTX-M, TEM, SHV, and GES, which are the most common lactamases, and Klebsiella pneumoniae (KPC) carbapenemases that can also hydrolyse carbapenems [94]. Mutational variability is a peculiarity of TEM and SHV β-lactamases. These mutations in the active site increase the volume of the enzyme and enable it to hydrolyze bulk molecules of second-to-fourth-generation cephalosporins. The mutant forms are known as extended-spectrum β-lactamases (ESBLs) [98].

Class C β-lactamases, which are effective in the hydrolysis of cephalosporins, are characterized by a broad spectrum of inactivation, including penicillins, early cephalosporins (e.g., cephalothin), and cephamycins (e.g., cefoxitin), along with a general lack of effect of inhibitory agents, such as clavulanic acid, sulbactam, and tazobactam [99].

Class D β-lactamases consist of fourteen families, of which the OXA family represents the largest and most heterogeneous. The enzymes belonging to the OXA family present a diversified spectrum of resistance, probably due to the variability of their active sites. This variability is also the basis for the distinction of class D β-lactamases from other classes of β-lactamases, although the hydrolysis profiles of some narrow-spectrum (NS) OXAs may appear similar to those of class A penicillinases [100].

Molecular class B is a heterogeneous family of metallo-β-lactamases (MBLs), which, unlike β-lactamases of classes A, C, and D which use a serine as the active center of the enzyme, uses metallic zinc ions (Zn^2+^) [101]. They show the ability to hydrolyze several β-lactam antibiotics, carbapenems, and cephalosporins but not those belonging to the monobactam class. In addition, the common commercially available inhibitors (clavulanic acid and tazobactam) are not able to block their catalytic activities; however, due to their zinc-containing active sites, they can be neutralized by metallochelants (EDTA, dipicolinic acid, and o-phenanthroline). The emergence of new MBL variants (e.g., NDM-type carbapenemases) and their co-expression with serine—β-lactamases represents a serious problem of antibiotic resistance, especially at a time when bacteria resistant to all β-lactam antibiotics are clinically relevant [102]. Another example of inactivation of the pharmacologically active molecule is represented by the production of macrolide esterases (MEs), which confer resistance to macrolides (erythromycin, azithromycin, etc.) These enzymes catalyze the hydrolysis of the lactone ring, preventing binding to the target site [67,80]. Among the esterases known to date, the maximum clinical importance is attributed to EreA and EreB, which are able to inactivate macrolides, such as erythromycin, via the hydrolysis cleavage of the macrolides’ macrolactone ring. EreA is a metal-dependent enzyme with a limited action profile. It does not hydrolyze azithromycin and telithromycin, and being a metalloenzyme, its activity is inhibited by chelating agents. However, among the components within the Ere family, EreA appears to be the enzyme identified with greater frequency in several clinical strains. Meanwhile, EreB confers resistance to almost all macrolides, except telithromycin [103]. The evolutionary development of bacteria through genomic mutations combined with the selection of new resistance phenotypes have led to the development of new antibiotic resistance enzymes, defined as bifunctional, i.e., encoded by two linked genes, which broaden the specificity for the substrate and the spectrum of resistance to different AMDs [104].

Table 2 overviews the main types of enzymes produced by bacteria, with respective enzyme reactions catalyzed and molecule target sites, responsible for inactivation of the pharmacologically active substance.

(ii)
*Antibiotic target site protection*


The targets of antimicrobial molecules used in clinical practice can be located within bacterial cells or, in the case of Gram-negative bacteria, in the cytoplasmic membrane. To exert the antimicrobial effect, it is therefore necessary for the compound to penetrate the external and/or cytoplasmic membrane. In view of this, bacteria have developed mechanisms to prevent the antibiotic from reaching its intracellular or periplasmic target through two routes: (a) decreasing penetration and (b) active expulsion of the antimicrobial compound (Table 3).

(a)As mentioned, there is an innate resistance in the ability of bacteria to limit the entry of antimicrobial agents into the cell. This natural difference is given by the cell wall, which in Gram-negative bacteria is quite complex, and provides a barrier to certain types of antimicrobial molecules [70]. In fact, it prevents the penetration of hydrophilic drugs, such as β-lactams, tetracyclines, and some fluoroquinolones and, therefore, their binding to the target site. In these bacteria, hydrophilic molecules are endocytosed through diffusion channels and porins [105,106]. However, drug uptake through porins can be limited by two ways: reduction in the number of porins and mutations [107]. Porin mutations could be achieved by three general processes, shift in the type of pore expressed, change in the level of pore expression, impairment of pore function. Antibiotic resistance, generated by any of these mechanisms responsible for changes in porin permeability, is often associated with other resistance mechanisms, such as increased expression of efflux pumps (described below) [41].(b)Efflux pumps are active transport proteins involved in the extrusion of substrates against a concentration gradient, including antibiotics, from inside to outside cells [41,108]. Since they rely on energy sources for the active transport of substances from inside to outside the cells, a first general classification divides them based on the mechanism by which they obtain this energy. Primary efflux pumps obtain energy from the active hydrolysis of ATP, while secondary efflux pumps obtain energy from chemical gradients formed by protons or ions such as sodium [109]. First described in the 1990s as a mechanism of drug resistance, numerous efflux pumps in bacteria have subsequently been characterized through molecular biology studies. Five major families of efflux pumps have been described in prokaryotes, namely: (i) adenosine triphosphate binding cassettes (ABCs), primary active transporters that utilize energy derived from ATP hydrolysis; (ii) small family of multidrug resistance (SMR family), unusually small proteins that are predicted to cross the membrane only four times; (iii) multidrug and toxin extrusion (MATE) family that uses a Na^+^ gradient as an energy source; (iv) major facilitator superfamily (MFS) that uses solute/cation (H^+^ or Na^+^) symporter or solute/H^+^ antiporter; (v) nodular resistance cell division (RND) family, capable of exploiting the substrate/H^+^ antiporter mechanism and involved in the efflux of multiple antibiotics simultaneously, being multidrug transporters, as well as molecules of detergents, dyes, heavy metals, solvents and many other substrates [70]. These families are classified according to the differences shown in their structural conformation, energy source, range of substrates they can extrude, and type of bacteria in which they are distributed.

(iii)
*Modifications and/or bypasses of target sites*


Another common strategy used by bacteria to develop antimicrobial resistance is to interfere with the target site through target protection (preventing the antibiotic from reaching its binding site) and through changes in the target site (point mutation, enzymatic alteration or replacement of original site), resulting in decreasing affinity for the antibiotic molecule.

Target protection is based on the physical association of a resistance protein (“target protection protein”) with the bacterial target site of the antibiotic to safeguard the latter from the action of the antibiotic. Such interaction between the target protection protein and the target must be persistent or repeated direct interaction [110]. Target protection was first recognized about 30 years ago as a mechanism of resistance to tetracycline and remained for a considerable period of time the only clearly documented example [111,112]. These proteins, known as TetM and TetO, were initially described in *Streptococcus* spp. and *Campylobacter jejuni*, but both have now been widely detected in several bacterial species, probably because they are transported by MGEs, plasmids, and broad-spectrum conjugative transposons [113]. TetM and TetO belong to the GTPase of translation factors and intervene in protein synthesis as homologs of elongation factors (EF-G and EF-Tu). The activity of these proteins is expressed at the ribosomal level in a GTP-dependent manner, through an interaction between domain IV of the 16S rRNA and the tetracycline-binding site that results in the removal of tetracycline from its binding site on the ribosome. Furthermore, this interaction by altering the ribosomal conformation prevents the subsequent binding of the molecule to the target site [114]. Furthermore, it has been demonstrated that these proteins are able to compete with greater binding affinity for the same ribosomal space with the aforementioned antibiotic and, at the same time, alter the geometry of the antibiotic binding site, preventing binding and allowing the resumption of protein synthesis [115]. Another example of target protection is the quinolone resistance protein (Qnr), mediated by plasmid mobile genetic elements, whose importance is linked to its frequent detection in clinical isolates [116]. It was first described in the mid-1990s in a clinical isolate of *K. pneumoniae* [117] and classified as belonging to the pentapeptide protein family. The mechanism of action used by this protein, which acts as a DNA homologue, involves competition for the DNA-binding site of DNA gyrase and topoisomerase IV, thus reducing the interaction of DNA with gyrase-DNA. Antibiotic resistance is thus given by a reduced availability for the quinolone molecule to form a DNA-quinolone complex cleaved by gyrase, which is lethal for the cell [116]. It is important to emphasize that this is a low-level quinolone resistance. However, the presence of genes encoding Qnr has been associated with the evolution of highly resistant bacteria by facilitating the selection of mutants with point mutations in genes encoding DNA gyrase and/or topoisomerase IV, which is the target site of the fluoroquinolone antibiotic class [118].

Regarding the modifications of the target site of antimicrobial compounds to decrease the affinity of antibiotic with target site, these may consist of (i) point mutation in the genes encoding the target site, (ii) enzymatic alterations of the binding site (e.g., addition of functional groups as methyl groups), and/or (iii) replacement or bypass of the original target.

Changes associated with the target site are often caused by spontaneous mutations in a bacterial gene on the chromosome as a result of environmental pressure in the presence of the antibiotic. Examples described in clinical isolates include mutations in RNA polymerase and DNA gyrase [119]. Mutations that achieve this result occur because organisms cannot completely relinquish vital cellular functions of target sites, so mutational shifts in the target of RNA polymerase and DNA gyrase occur, reducing susceptibility to inhibition while maintaining cellular function [119]. One of the most classical examples of mutational resistance is the development of rifampicin (RIF) resistance. The mechanism of action of this antibiotic, belonging to the rifamycin class, consists of blocking bacterial transcription by inhibiting the DNA-dependent RNA polymerase, directly blocking the path of nascent RNA [120]. It is noteworthy that although the single-step point mutations responsible for RIF resistance give rise to high-level resistance, these mutations result in a decrease in the affinity of the drug for its target but usually spare the catalytic activity of the polymerase, maintaining transcription unaltered [121].

Another example that well represents antibiotic resistance through target site modifications is the development by bacteria of chromosomal mutations in the genes encoding the subunits of DNA gyrase and topoisomerase IV, which represent the binding site for fluoroquinolones (FQs). Both enzymes (DNA gyrase and topoisomerase) are essential for bacterial survival, so the level of resistance achieved following mutations in one of the enzymes will depend on the ability of the antibiotic to effectively inhibit the unchanged target. This dual interaction of FQs with two distinct sites of action may explain a minor increase in MIC values in the early stages of resistance development, unlike mutational resistance acquired for RIF or other antibiotic molecules [122].

Finally, a further example of resistance resulting from mutational changes is resistance to oxazolidinones (linezolid and tedizolid), a synthetic antibiotic that acts at the ribosomal level by interfering with the positioning of aminoacyl-tRNA at the A site, inhibiting amino acid coding and consequently protein synthesis, with a bacteriostatic effect on aerobic and anaerobic Gram-positive bacteria. The most representative antibiotic of this class in clinical therapy is linezolid. The mechanisms of resistance to this antibiotic are best characterized to date, including mutations of the genes encoding for the V domain of 23S rRNA and/or the ribosomal proteins L3 and L4 (rplC and rplD, respectively) and methylation of 23S ribosomal RNA in the 23S rRNA mediated by the enzyme Cfr methyl transferase [123]. Although the resistance to it currently restricted does not cause alarm, it is important to keep in mind that, in the majority of clinically relevant Gram-positive organisms, it has been detected [123].

Target modification is an increasingly popular resistance strategy among pathogens. Resistance to glycopeptide and polymyxin antibiotics, molecules capable of interfering with the formation of the bacterial cell wall, is operated by the activity of enzymes that chemically alter the terminal units of the cell wall precursors (D-alanyl-D-alanine) that represent the antibiotic binding site. Another emerging class of target-modifying enzymes are methyl transferases that, through the methylation of ribosomal subunits, reduce bacte-rial sensitivity to aminoglycoside antibiotics, lincosamide, streptogramin, macrolides, and oxazolidinone [124]. A well-documented instance of resistance by enzymatic modification is the methylation, either mono- or di- methylation, of the adenine residue at position A2058 in the V-domain of the 23S rRNA of the 50S ribosomal subunit, resulting in resistance to macrolides. Due to the overlapping binding sites in the 23S rRNA, macrolides, lincosamides, and streptogramin B antibiotics (together known as the MLSB group) exhibit cross-resistance to all members of this group [125].

Several modes of complete modification or bypass of the target site have been documented [126]. A well-known mechanism is that generated by the replacement of normal penicillin-binding proteins (PBPs) with PBP2a, which, by reducing the affinity for β-lactams, confers resistance to this class of antibiotics. The induction of this mutation occurs as a result of the environmental pressure exerted by the presence of β-lactams [33,126]. Another notable example is related to the acquisition of a biochemical mechanism capable of conferring resistance to glycopeptides (e.g., vancomycin and teicoplanin). This resistance is determined by structural mutations of the peptidoglycan during its synthesis, such as: (i) modification of the last D-Ala with D-lactate (high-level resistance) or D-serine (low-level resistance); (ii) destroying the “normal” D-Ala-D-Ala terminal cell wall precursors, to reduce the affinity of vancomycin antibiotics to peptidoglycan precursors [127].

An additional “target bypass” strategy involves increasing the production of the available antimicrobial targets with the aim of overpowering the antibiotic, as in the case of TMP-SMX resistance. This drug works by altering the biochemical mechanism for folate synthesis impairing bacterial synthesis. The folate synthetic pathway involves two major enzymes, dihydropteroate synthase (DHPS), which forms dihydrofolate from para-aminobenzoic acid (inhibited by SMX), and dihydrofolate reductase (DHFR) which catalyzes the formation of tetrahydrofolate from dihydrofolate (inhibited by TMP). Resistance to TMP-SMX can evolve through several mechanisms, among which are documented the reduction in the binding affinity between antibiotic molecules and enzymes, “target modification” and acquisition of external genes that encode for the production of isoenzymes (DHPS or DHFR) less sensitive to inhibition by TMP/SMX, “target bypass”, another “genetically based” bypass strategy occurs through mutations in the promoter region of the genome encoding these enzymes that induce an overproduction of DHFR or DHPS [128]. Overproduction of these enzymes over-saturates the ability of TMP/SMX to inhibit folate production and allow bacterial survival [129]. Bacteria such as enterococci use another “bypass” strategy: developing the ability to use folate from multiple sources, incorporating tetrahydrofolic acid and exogenous folic acid when weighing in the external environment [130,131]. This strategy is correlated with strong increase in TMP-SMX MICs and is thought to compromise antimicrobial activity in vivo [132].

Table 4 summarizes main strategies relying on target site modifications and bypass mechanisms for bacterial antimicrobial resistance.

(iv)
*Global cellular adaptive process*


Over time, bacteria have developed various evolutionary mechanisms to survive in the external environment (defending itself against other microorganisms, including competing bacteria and bacteriophages) and counteract the pressure of the immune system in the host organism [133]. A clinically relevant example is the adaptive evolution of daptomycin (DAP)- and vancomycin (VCM)-resistant phenotypes, which represents the expression of the bacterial cell’s response to antibacterial challenge [134]. DAP is a bactericidal lipopeptide antibiotic related to cationic antimicrobial peptides (CAMPs) produced by the innate immune system, which induces important electrochemical, structural, and functional alterations, thus altering the homeostasis of the bacterial cell up to death without cell lysis [135]. In response to the action of CAMPs, bacteria have developed defense systems involved in the protection of the cell membrane when attacked by CAMPs [136]. Population analysis of vancomycin-resistant strains has revealed the presence of subpopulations of bacterial cells exhibiting MICs between 4 and 8 μg/mL known as VISA isolates and MICs above 2 μg/mL (clinical breakpoint for susceptibility) designated as the phenotype heterogeneous VISA (hVISA) [137].

The specific mechanisms leading to the hVISA/VISA phenotype are not yet fully known. However, prolonged exposure to glycopeptides predisposes to the development of ordered and sequential genetic mutations of systems involved in the control of cellular homeostasis, including the cell wall and genetic regulatory proteins, such as *asem*A,B,C,X; *pbp*A, *pbp*B,C,D; *vra*SR) that result in essential cell wall remodeling in the hVISA/VISA phenotype [138]. Finally, an interesting feature is the remarkable genetic plasticity of many hVISA/VISA strains in the absence of vancomycin exposure, capable of switching from a sensitive (or even completely susceptible to vancomycin) to a resistant phenotype, highlighting the bacteria’s high ability to evolve in response to environmental stresses [138].

Another defense mechanism observed in the bacterial community, especially during colonization phases, is the formation of biofilms. In this polysaccharide matrix, one predominant organism may be present (such as *Staphylococcus* spp., *Corynebacterium* spp., and *Propionibacterium* spp.), or they may be composed of a wide variety of organisms. Following the production of the biofilm, other planktonic bacteria may be recruited, making the newly formed structure increasingly complex [139]. Biofilm formation, due of the dense and sticky consistency of the polysaccharide matrix, protects bacteria from attack by the host’s immune system, as well as providing protection from antimicrobial molecules, which require much higher concentrations to be effective. Furthermore, bacterial cells in biofilms tend to be sessile (slow rate of metabolism, slow cell division), so antimicrobials targeting cell replication mechanisms have limited effect. An important consideration in biofilms is the ability of bacteria to communicate and transfer genetic material, an event facilitated by the proximity and promiscuity of bacterial cells [140].

## 6. Classification and Evolution of Resistance Genes

The widespread use of antibiotics in clinical practice and prophylaxis generates selective pressure on bacterial populations, which induces cells to develop and maintain antibiotic resistance mechanisms. As previously said, the prevailing belief is that the issue is only linked to the use and misuse of antibiotics in people and animals. This applies to the clonal dissemination of pathogenic bacteria exhibiting resistance mechanisms that arise from alterations in target molecules, resulting from mutational events and robust positive selection of mutants. Nonetheless, the majority of antibiotic resistances are probably instances of acquired resistance, resulting from the lateral transfer of antibiotic resistance genes (ARGs) between ecologically and taxonomically disparate bacteria [6,7]. This leads to the emergence of resistant strains capable of developing resistance to various antibiotics [141]. Resistance genes can be categorized according to the specific class of antibiotics to which they provide resistance, including cyclines as tetracyclines (*tet*), sulphonamides (*sul*), β-lactams (*bla*), macrolides (*erm*, *mrs*, *ere*), aminoglycosides (*aac*, *ant*, *aph*), quinolones (*aac*, *gyr*, *par*, *qnr*), peptide (*mcr*), glycopeptides (*van*), and multidrug resistance [142].

Tetracycline antibiotics. The first tetracycline antibiotic was produced in 1948 by ex-traction from *Streptomyces aureofaciens* and characterized as chlortetracycline from *Streptomyces* spp. [143]. In the following decades, additional tetracyclines were identified primarily in *Streptomyces* spp. (e.g., oxytetracycline, tetracycline) or products of semi-synthetic approaches (e.g., doxycycline and minocycline) [143]. Resistance to tetracyclines has developed prematurely, just think that until the mid-1950s, most bacteria, including pathogenic ones, were sensitive to tetracycline, but the first tetracycline-resistant bacteria were isolated as early as 1953 [144]. Tetracycline resistance mechanisms include three key strategies: energy-dependent efflux pumps (ABC efflux pumps), ribosomal protection proteins (RPPs, TetM and TetO) or enzymatic inactivation (TetX) [145].

Sulfonamide antibiotics. Sulfonamides belong to the oldest synthetic drugs, first introduced in 1932 [146], with sulfamethoxazole in combination with trimethoprim being the most widely used currently [147].

Resistance to sulfonamide arises from mutations in the *fol*P gene, which encodes dihydropteroate synthase (DHPS) [148]. Resistance to sulfonamides was recognized in the 1960s, and the plasmid-mediated genes *sul*1 and *sul*2 were characterized post-1980s [149]. Furthermore, a third plasmid-mediated *sul*3 gene was identified [150].

Beta-lactam antibiotics. Antibiotics belonging to the β-lactam family have a β-lactam core in their molecular structure and include penicillins and derivatives, cephalosporins, carbapenems, monobactams and β-lactam inhibitors [151]. Resistance to β-lactams can be developed through several mechanisms, including the production of β-lactamases, such as extended-spectrum β-lactamases (ESBLs), ESBL genes (*bla*CTX-M, *bla*SHV, *bla*TEM), *Amp*C-mediated enzymes from plasma and the β-lactamases that hydrolyze carbapenems (carbapenemases). Currently, more than 1.150 β-chromosomal lactamases, plasmids, and transposons are known [96,152]. However, there are some bacteria, such as *Stenotrophomonas maltophilia*, that have an endogenous metallic β-lactamase (MBL) L1 that makes them resistant to carbapenems [153]. Some mechanisms of resistance are common between Gram-positive and Gram-negative bacteria, such as lowering drug susceptibility by reducing the affinity of the target drug or reducing drug penetration. For instance, among Gram-positive bacteria, carbapenemase resistance is acquired through mutations in penicillin-binding proteins (PBPs) [154]. Similar to ampicillin resistance, *Haemophilus influenzae* reduces affinity for macrolide targets through modifications in 23S rRNA or ribosomal proteins L4 and L22, leading to high levels of resistance [155]). While resistance attributed to lower drug penetration is achieved through mutations and/or downregulation of porin expression, or active extrusion by efflux pumps [156]. In contrast to Gram-positive bacteria, PBPs are located in the periplasmic space of Gram-negative bacteria and β-lactam antibiotics must cross through the bacterial outer membrane to reach their target action sites. In Gram-negative bacteria, the resistance attributed to decreased drug penetration is obtained through reduced expression of the pores of the outer membrane, e.g., OmpF, OmpC e phoe in *E. coli* e OprD in Pseudomonas aeruginosa, and to expression of Bradly’s multidrug-specific efflux system (Mex), such as NexAB-OprM and Mex-XY-OprM in Pseudomonas aeruginosa [157]. The flow pumps, found in both types of bacteria, are more important for Gram-negative bacteria to extrude β-lactam antibiotics and often work together with other resistance mechanisms, as β-lactamases, to increase their resistance to antibiotics [158]. Regarding the enzymes responsible for carbapenem resistance, the main efficient carbapenemases responsible for carbapenem hydrolysis are the KPC, VIM, IMP, NDM and OXA-48 types [159,160,161].

Macrolide antibiotics. The first macrolide, erythromycin A, was discovered in the early 1950s [162], and new macrolides such as clarithromycin and the azalide, azithromycin, were subsequently developed [54]. Shortly after the introduction of erythromycin in the clinical setting, bacterial resistance to this antibiotic was first reported in staphylococcal isolates [163]. Since then, several genes encoding macrolide resistance have been characterized. Many bacteria harbor the AR genes responsible, including rRNA methylases, efflux pumps and inactivation genes divided into esterases, lyases, phosphorylases and transferases [54].

The predominant resistance mechanism arises from rRNA methylases, encoded by the *erm* genes, whilst the alternative mechanisms, efflux pumps and inactivating genes, are represented by the *msr* and *ere* determinants, respectively [54].

Aminoglycoside antibiotics. The first aminoglycoside discovered was streptomycin in *Streptomyces griseus* in the early 1940s [164]. Several years later, other aminoglycosides were characterized from other *Streptomyces* species, such as gentamycin, tobramycin and amikacin from *Micromonospora purpurea*, and kanamycin from *Streptomyces tenebrarius*. In the 1970s, the first semi-synthetic derivatives such as netilmicin and amikacin were produced [165]. The main mechanism of resistance to aminoglycosides involves modification of enzymes (AMEs, aminoglycoside modifying enzymes).

Three primary kinds of proteins, encoded by the AME genes *aac*, *ant*, and *aph*, have been categorized according to the type of modification. (i) AAC (acetyltransferases), comprising AAC(1), AAC(2), AAC(3), and AAC(6); (ii) ANT (nucleotide transferase or adenyl transferase), encompassing five nucleotidyl transferases: ANT(2), ANT(3), ANT(4), ANT(6), and ANT(9); (iii) APH (phosphotransferase), consisting of seven phosphotransferases: APH(2), APH(3), APH(4), APH(6), APH(7), and APH(9) [74,164].

Quinolones antibiotics. The discovery of quinolone molecules dates back to 1962, when during the synthesis and purification process of chloroquine (an antimalarial agent), a quinolone derivative, nalidixic acid, was discovered with bactericidal activity against Gram-negative bacteria [149]. Subsequently, by adding a fluorine atom to the 6 positions of a quinolone molecule, fluoroquinolones, also known as second-generation quinolones, were generated with a broader and more efficient spectrum of action. During the 1980s, various fluoroquinolones were developed, for example, ciprofloxacin, norfloxacin and ofloxacin, broad spectrum effective against both Gram-positive and Gram-negative bacteria. During the 1990s, further changes led to the third generation (fluor) quinolones, such as levofloxacin and sparfloxacin, trovafloxacin, with an encouraging also active against anaerobic bacteria [166,167]. Genes encoding mechanisms of bacterial resistance to quinolones are mainly located in chromosomes, and plasmids capable of carrying some resistance mechanisms have been identified [117,168]. Chromosome-related resistance mechanisms include reduced membrane porin density to decrease the ability of the antibiotic to enter the cell or overexpression of efflux pumps to increase exocytosis, and mutations in quinolone resistance determining regions (QRDRs) [166,169]. The latter have been identified in the genes *gyr*A, *gyr*B, *par*C and *par*E which synthesize the subunits of DNA gyrase and topoisomerase IV. Regarding plasmid-mediated resistance, the first quinolone resistance gene detected was qnr, of which five *qnr* gene lineages have been subsequently identified: *qnr*A, qnrB, *qnr*C, *qnr*D and *qnr*S. Other plasmid-mediated quinolone resistance genes are a cr variant of *aac*(6)-Ib, namely *aac*(6)-Ib-cr, which encodes the aminoglycoside acetyltransferase [170] and the *qep*A gene, which encodes an efflux pump capable of excreting hydrophilic fluoroquinolones, e.g., ciprofloxacin [171,172].

Colistin antibiotic. Colistin (also known as polymyxin E) was one of the first antibiotics with significant activity against Gram-negative bacteria, particularly *Pseudomonas aeruginosa*, against which it showed rapid and concentration-dependent bactericidal activity [173,174]. This drug, marketed for fifty years as the inactive prodrug colistin methanesulfonate (CMS) [175], starting from the years’70, following the reported nephrotoxicity and neurotoxicity, it has been largely replaced by aminoglycosides [160]. The infrequent use of these antibiotics could explain the relatively low resistance rates; however, several outbreaks of infections caused by bacteria resistant to colistin have been recorded [176,177]. The molecular mechanisms associated with colistin resistance in Gram-negative bacteria are diverse, including changes with the two-component systems PmrA/PmrB, PhoP/PhoQ, ParR/ParS, ColR/ColS, and CprR/CprS and alterations in the *mgr*B gene, which codes for negative regulator of PhoPQ. These alterations result in the loss of the molecule’s target through the neutralization of lipid A biosynthesis with total loss of LPS [81]. Until a few years ago, polymyxins remained one of the last classes of antibiotics in which horizontal, plasmid-mediated transmission had not been recorded, whereas today the possibility of plasmid transmission is well known [178,179,180]. This resistance arises from mutations (small nucleotide changes) in the *mcr*-1 gene, which lead to, giving rise to gene variants, from *mcr*-1.2 to *mcr*-1.27 [89,180]. It has recently been known that there has been an increase in resistance to polymyxin worldwide due to its increased use in clinical practice and, given the current situation, this raises several concerns [181]. Given the current situation, this raises several concerns, and it therefore appears necessary to deepen the current knowledge on polymyxin resistance through standardized and rapid methods, such as genotyping, in addition to common in vitro polymyxin susceptibility tests, as discussed by Rubens et al. [181].

Glycopeptide antibiotics. In the late 1950s, the glycopeptide antibiotic used in clinical settings was the vancomycin, a glycopeptide isolated as a fermentation product from *Streptomyces orientalis*, a telluric bacterium [162]. Currently, four groups of glycopeptides are recognized: vancomycin-type, avoparcin-type, ristocetin-type, and teicoplanin-type [182]. Among these, vancomycin and teicoplanin are the only two molecules currently used against Gram-positive microorganisms. The molecular target site of these glycopeptide antibiotics is located on the peptidoglycan precursor of the cell wall, specifically the D-alanyl-D-alanine (D-Ala-D-Ala) terminus, where they bind to inhibit the subsequent trans-glycosylation reaction by competitive inhibition [183,184]. In the 1990s, an association was demonstrated between the use of avoparcin and the emergence of glycopeptide-resistant enterococci (GRE), more commonly known as vancomycin-resistant enterococci (VRE), in farm animals [185,186]. Based on this report, the use of avoparcin as a growth promoter has been banned in all European Union countries since 1997. However, resistance to glycopeptides, such as vancomycin and teicoplanin, or both, has been detected in six Gram-positive bacteria: *Enterococcus* spp., *Erysipelothrix* spp., *Lactobacillus* spp., *Leuconostoc* spp., *Pediococcus* spp. and *Staphylococcus* spp. [187].

Vancomycin resistance is linked to gene activation *van*A, *van*B and *van*D which give rise to the production of the modified peptidoglycan precursor, D-Ala-D-Lac, and vanC, vanE and vanG that modify the D-Ala-D-Ser precursor, for which the glycopeptides show a reduction in binding affinity. The localization of the *van*A and *van*B operons has been documented on both plasmids and chromosomes; while *van*C1, *van*C2/3, *van*D, *van*E and *van*G appear to be localized exclusively on chromosomes [184].

Multidrug resistance. The expression of multidrug-resistant bacterial multidrug efflux systems is usually controlled by transcriptional regulators, which inhibit or activate the transcription of multidrug efflux genes [188] and is associated with mutations in intergenic sites that control the expression of these genes (transcription factors) or their regulators (repressor genes) [35,189,190]. A single base pair mutation was identified in the consensus sequence of the *mtr*C gene in *Neisseria gonorrhoeae*, leading to the production of a novel and more potent promoter that caused the overexpression of efflux pumps and, thus, multidrug resistance [35]. Another mechanism that favors the development of MDR is the ability of bacterial communities to form biofilms, thus limiting the action of antibiotics and the immune system, while facilitating the transmission of resistance genes between pathogenic microorganisms with consequent increase in bacterial virulence and development of multi-resistant phenotypes [191,192]. In addition, the transmission of resistance can involve commensal bacteria in multi-resistant pathogens [193].

A resistome analysis of data published between 1990 and 2020 revealed that the predominant Antimicrobial Resistance Genes (ARGs) isolated from hospitals were multidrug, glycopeptide, and β-lactam ARGs (*mec*A, *van*A, *van*B, and *bla*), whereas sulfonamide and tetracycline ARGs (*sul* and *tet*) were most prevalent in farms, wastewater treatment plants (WWTPs), as well as in water and soil. Within six habitat categories (farms, cities, WTP, water, soil, and air), the leading 50 topic subtypes included eight families of antibiotics: β-lactam (*bla*), sulfanilamide (*sul*), tetracycline (*tet*), aminoglycoside (*aad*), multidrug (*mec*), amphenicol (*flo*), trimethoprim (*dfr*), and glycopeptide (*van*). The most prevalent among them globally was *mec*A (multidrug), followed by β-lactams (*bla*), glycopeptides (*van*A, *van*B), sulfonamides (*sul*1, *sul*2), and tetracyclines (*tet*M). The ten most prevalent ARGs identified in Asia were *bla*NDM-1, *bla*CTX-M-15, mecA, *bla*TEM-1, *sul*1, *van*A, *bla*KPC-2, *sul*2, *bla*CTX-M-14, and *bla*OXA-48, associated with β-lactam, multidrug, sulphonamide, and glycopeptide antibiotics. This perspective clearly indicates that the dissemination of ARGs is a worldwide issue devoid of borders [194].

## 7. Development of Resistance Genes in Different Environments and the Part Played by Humans

Agricultural and soil environments can be considered as confluence basins between different ecosystems, which favors the mixing and diffusion of resistance genes. The presence of these “reservoirs” of ARGs can represent a serious risk for human and animal health due to the close interactions that can exist between humans and the agricultural environment [192]. Worldwide, animal agriculture constitutes over fifty percent of overall antibiotic use, estimated at 131,109 tons in 2013, with projections exceeding 200,000 tons by 2030 [195]. These figures are affected by the use of prophylactic antibiotics in countries where their administration is authorized. In 2019, about 17 million kg of antimicrobials supplied in the United States were designated for food-producing animals, with more than one-third of this quantity including antimicrobials critical to human health (FDA, 2020). Furthermore, it is projected that BRICS nations (Brazil, Russia, India, China, and South Africa) would quadruple their antimicrobial use in livestock by 2030, with India’s application of antimicrobials in poultry anticipated to treble by that year [196]. All this confirms that close human–animal–environment interactions, through the food chain, or through direct contact between animals and humans, can promote the exchange of mobile ARGs [197,198,199]. Antibiotic treatment and/or ingestion of foods with antibiotic residues or contaminated (fecal or soil bacteria) foods can favor the integration of antibiotic-resistant bacteria (ARBs) and the increase in ARGs in the intestinal microbiota. For example, selective pressure from residual antibiotics (e.g., >60 ng/kg in meat products) alters the type and increases the number of ARGs in the human gut [200]. Newly formed ARBs can infect humans via various routes [201,202]. In fact, it has been demonstrated, through molecular biology techniques (genomic sequencing), the presence of *bla*NDM-positive *E. coli* in livestock, as well as in flies, dogs, and farmers, providing tangible evidence of the potential diffusion of the *bla*NDM gene and the possibility of carbapenem-resistant *Escherichia coli* to rapidly contaminate humans via dogs, flies, and wild birds [203,204].

ARBs and ARGs from organic fertilizers have been detected in crops grown on soils enriched with organic fertilizers [205]. Endophytic bacteria possessing ARGs can colonize plants and persist during the plant growth phase [206].

Microbial ecosystems are therefore not isolated, and it has been demonstrated that mobile ARGs shared between humans and animals have been identified in the intestine of humans, chickens and pigs, which confer resistance to six main classes of antibiotics: tetracline, aminoglycoside, macrolide-lincosamide-streptogramin B (MLSB), chloramphenic, β-lactamic, and sulphonamide [199]. Antimicrobial resistance on a farm also impacts antimicrobial resistance in the environment and on human and animal health. It has been shown that tetracycline applied to pigs is rapidly excreted, and up to72% of the antibiotic is excreted in active form in the feces and urine of treated animals [207]. This supports the possibility of transmission of new resistances to the soil microbiota, which naturally hosts ARGs, known as intrinsic resistome [208]. In addition, the spreading of feces and urine of grazing animals and the application of animal manure as fertilizer can generate a selective pressure generated by the elimination of the active ingredient in the ecosystem and contribute to the spread of multi-resistant bacteria in the soil [209,210]. For example, a study documenting the consistency of antibiotic-resistant reservoir microorganisms in large-scale pig farms in China highlighted the presence of resistance genes for all major classes of antibiotics, including antibiotics of critical medical importance to humans, such as macrolides (*mpa* and *erm* genes), cephalosporins (*bla*TEM and *bla*CTX-M), aminoglycosides (*aph* and *aad* genes), and tetracyclines (*tet* genes) [211].

Many studies have been conducted on tetracycline resistance, demonstrating the presence of tetracycline-resistant genes in bacteria isolated from soil and pig manure [212]. Among the most common genes isolated from soil were *tet*L, *tet*A, *tet*M, *tet*W, and *tet*G [209], while in manure there were *tet*Q, *tet*W, *tet*X, *tet*O, *tet*M, *tet*L, and *tet*G [211]. Consequently, common manure management practices, such as composting, and subsequent application to the soil can influence and favor the spread of mobile elements of antibiotic resistance. Soil can also be contaminated with antimicrobials, ARGs, and resistant organisms using sewage sludge for agricultural production and irrigation wastewater from industrial, agricultural, pharmaceutical, and municipal treatment plants [213]. In 2017, Wang et al. [214] demonstrated that the chicken intestine has more MGE than that of humans and pigs; furthermore, considering that poultry litter is also used as a fertilizer or energy source, it represents a source of diffusion of the ARGs. The results of some studies conducted on the presence of ARGs in the excrement of chickens from poultry farms show the presence of resistance genes to tetracycline (*tet*Q and *tet*M) and to sulphonamides (*sul*2), as well as resistance to β-lactams (*bla*OXA-1, *bla*MOX-like, *bla*CIT-like, *bla*SHV, and *bla*FOX) is ubiquitous in livestock farms [215]. The possible spread of pathogens in poultry farms up to 3 km away through aeration systems should also be highlighted [216].

Many studies on ARGs have also focused on other environments, such as wastewater treatment plants (WWTPs) [217]. Antibiotic resistance genes detected in environmental phage DNA showed the presence of ARGs for all major classes of antibiotics. Regarding β-lactams, the genes *bla*CTX-M-25, *bla*TEM-116,229, *bla*TEM, *bla*CTX-M, *bla*PSE, *bla*CMY-2, *bla*KPC, *bla*OXA-48-like, and *bla*NDM are described in the WWTPs [218,219]. ARGs for quinolones are found: *qnr*A, *qnr*S in human fecal samples, *aac* (6′)-Iu, Is, Iy, *qnr*A3 in urban surface water, and *aac* (6′)-Ib-cr in river water [218,220,221]. VanY is the glycopeptide resistance gene detected in urban surface water [220]. Sulfanilamide resistance genes, *dfr*B2 and *sul*1, are isolated in surface water and seawater, respectively [220,222]. The macrolide resistance gene, encoded by *erm*F, was described in river water by [223]. While *tet*W, a tetracycline resistance gene, has been found in the Antarctic and Mediterranean environment [222].

ARGs identified in aircraft wastewater have a high diversity compared to those identified in conventional sewage, and, in particular, the *bla*CARB-4 β-lactamase gene was relatively abundant, representing a further possible element of rapid and global spread of resistance to antibiotics [224].

Environments contaminated by emissions from pharmaceutical manufacturing facilities and smog had the greatest relative abundance and variety of antibiotic resistance genes (ARGs) [225]. Biofilms present in rivers, streams, hot springs, and on human and animal teeth may enhance antibiotic resistance [226]. Microplastics (MPs) present in water are effective vectors of environmental microorganisms and antibiotic-resistant bacteria. The hydrophobic surface of plastic provides a refuge for microorganisms; in fact, it favors the formation of biofilms, facilitating the interaction between microorganisms and the environment. Furthermore, MPs modify the composition of ARGs in water and sediments, and, in turn, mobile genetic elements promote their diffusion [227].

Ultimately, research indicates that the association between ARGs and the mobile genetic element Inti1 is diminished in rural regions compared to urban and industrial locales [228]. In the opinion of a study conducted by Li and collaborators in 2018 that evaluated the presence of antibiotic resistance genes linked to total environmental particulate matter in the urban atmosphere of 19 cities in 13 countries around the world through high-throughput molecular biotechnology, the global atmosphere is contaminated to varying degrees by ARGs, highlighting the potential threat of airborne transmission of MGEs and the need to redefine current air quality standards to reduce the risk to public health [229].

## 8. Hypotheses for Solutions to Antibiotic Resistance

The rapid emergence of drug-resistant bacteria is plaguing the world due to the overuse and misuse of these drugs, as well as the lack of development of new molecules because of reduced investments in healthcare and challenging regulatory requirements [22,230]. In addition, microbes have acquired resistance even against antibiotics as a last resort. The World Health Organization (WHO) maintains a worldwide priority list of antibiotic-resistant bacteria, categorizing diseases by species and resistance type, and then classifying them into three priority levels: critical, high, and medium [231]. *Mycobacteria* (including *Micobacteriu tuberculosis*) are not part of this list, but extensively drug-resistant tuberculosis strains (XDR-TB) are noteworthy because they appear resistant to most drugs, including isoniazid and rifampin, all fluoroquinolones, and any of the three second-line injectable drugs (i.e., amikacin, kanamycin, and capreomycin). Consequently, drug-resistant mycobacterial infections and XDR-TB pose a growing threat worldwide, and new methods of treatment for tuberculosis are urgently needed [232].

Twenty years have elapsed since the WHO released its first “Global Strategy to Contain Antimicrobial Resistance (AMR)”, cautioning that AMR poses a significant danger to both animal and human health, in addition to the global economy [233,234]. To date, surveillance based on a “One Health” approach is essential to effectively assess the spread of antimicrobial resistance [198]. A first step to regulate the use of antibiotics and slow down the development of resistant bacteria is to plan targeted information days aimed at primary healthcare workers and mass media education campaigns aimed at the public on the use of antimicrobials only after investigations specifications [230].

To contain the transmission of ARGs from animal farms, hospitals and wastewater treatment plants, into the environment, the focus is shifting to biological technologies. Effective methods for the treatment of organic waste, such as manure and livestock sludge, useful for reducing ARGs include aerobic composting, anaerobic fermentation, biological treatment, UV disinfection and chlorination. Among these, aerobic composting seems to be one of the most efficient approaches to remove resistance genes. In fact, it has been shown to be effective in inactivating tetracycline resistance genes (*tet*Q, *tet*W, *tet*C, *tet*G, *tet*Z and *tet*Y), sulfanilamide resistance genes (*sul*1, *sul*2, *dfr*A1 and *dfr*A7) and fluoroquinolone resistance (*gyr*A) [235]. Solid-state anaerobic fermentation (SAD) is another effective method to remove ARGs and MGEs (such as plasmids, integrins, and transposases) under different conditions than conventional liquid anaerobic digestion (AD) [236]. Compared to conventional liquid AD treatment, SAD treatment significantly reduced the abundance of six ARGs, including *tet*C, *sul*2, *erm*Q, *erm*X, *qnr*A, and *aac* (6′)-ib-cr. Optimization of AD parameters, such as temperature and solid retention time, can positively influence the microbial community constitution and significantly improve the bio-degradation capacity of fecal pollutants [236,237]. To reduce ARGs in wastewater and drinking water sludge, some studies have analyzed water treatment processes. In the artificial treatment system, an example can be represented by the primary sedimentation tank of municipal wastewater treatment plants, through which it is possible to reduce the density of tetracycline resistance genes *tet*A and *tet*B [238]. Another method can be represented by UV disinfection, which has shown the ability to significantly reduce the abundance of *mec*A (reduction of about 1 log), even if it seems ineffective for the *van*A gene [239]. However, it should be considered that although disinfection treatment can reduce the total amount of resistant bacteria, it can also determine a selection of resistant bacteria, leading to a percentage increase in some AMRs [240]. Phage therapy is another frontier in the prevention of the production of resistance genes and in the treatment of drug-resistant bacterial infections, also due to the lack of harmful effects on mammalian cells [241,242].

The co-production of “synergistic antibiotics”, or “hybrid antibiotics”, with enhanced bioactivity is significant in addressing bacterial resistance [243] Examples of biosynthetic superclusters include the coproduction of sulfazecin and bulgecin in *Paraburkholderia acidophila* ATCC 31363 and *Burkholderia ubonensis* ATCC 31433 [244] and cephalomycin–clavulanate biosynthetic gene clusters (BGCs) in *Streptomyces clavuligerus* [152]. The bactericidal properties of sulfazecin and bulgecin exhibit synergy, with bulgecin shown to augment the antibacterial efficacy of clinical β-lactam antibiotics, such as third-generation cephalosporins, ceftazidime, and meropenem. The coproduction of cephalomycin with clavulanate, both β-lactams, is beneficial because cephalomycin, a cephalosporin antibiotic, inhibits bacterial penicillin-binding proteins and is susceptible to certain β-lactamases, while clavulanic acid effectively inhibits many β-lactamases [152,244].

Research has also investigated the potential use of natural substances, such as essential oils (EOs), to counter the antibiotic resistance issue [245,246,247,248]. Indeed, EOs, both alone and in EO-antibiotic combinations, have been shown to positively modulate the susceptibility of antibiotic-resistant pathogens even when they are organized in biofilms [247,248]. Indeed, EOs and the EO–antibiotic combination represents a promising therapeutic strategy against antibiotic-resistant bacteria, even protected within biofilms, which could allow decreasing concentrations of antibiotics used (personal observations).

## 9. Conclusions

Despite the attention paid in recent decades to the problem of AMR and the risks of the evolution of bacterial resistance mechanisms, until an integrated approach of collaboration between professionals from different sectors is used in a One Health perspective, the problem will not be solved of antibiotic resistance. The salient points for the control of bacterial resistance are as follows: (i) therapeutic use of bacteriophages, (ii) promoting active immunization through the use of vaccines, (iii) co-production of synergistic antibiotics, i.e., hybrid antibiotics with greater bioactivity or their use in combination with natural antimicrobial substances (OE), (iv) information/training of healthcare personnel for an informed use of antibiotics and on the health risks of the spread of AMR [18].

## Figures and Tables

**Figure 1 antibiotics-14-00222-f001:**
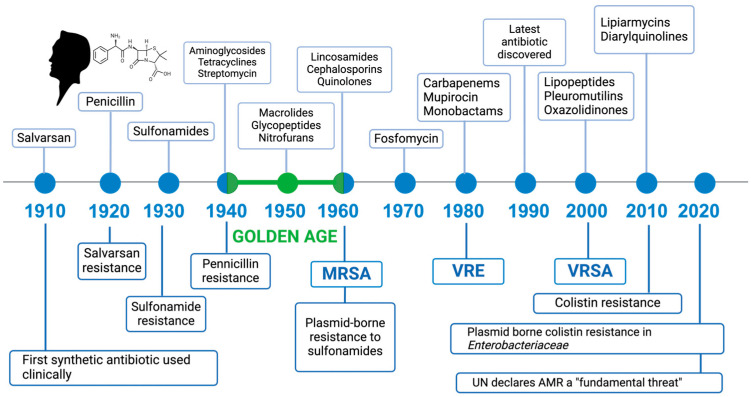
Historical progression of antibiotic use and the consequent development of antibiotic resistance phenomena.

**Figure 2 antibiotics-14-00222-f002:**
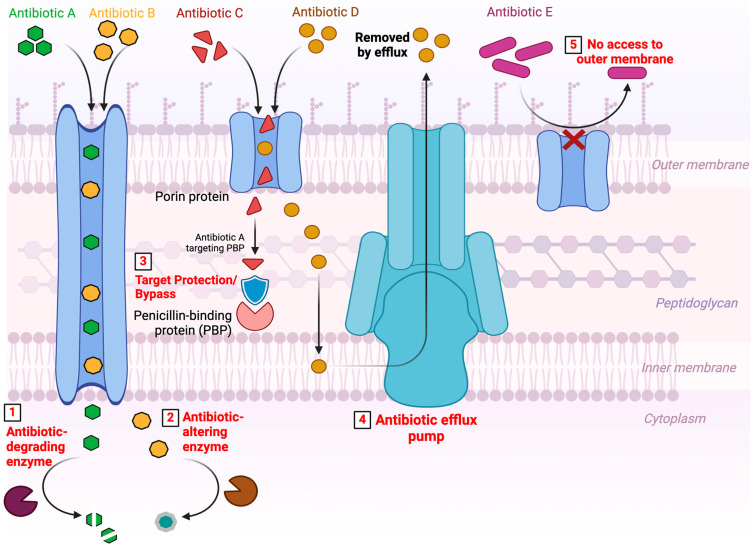
Graphical representation of the principal antimicrobial resistance mechanisms.

**Table 1 antibiotics-14-00222-t001:** Antibiotic action mechanism.

Antibiotic Targets	Antibiotic Action Mechanism	Antibiotic Class and Major Molecules
Cell wall	**Bactericidal agents** limit the formation of peptidoglycan, disrupt peptidoglycan cross-linkage, and impede precursor movement.	**Glycopeptide**: Vancomycin Bacitracin **Beta-lactam**: Penicillin, cephalosporin (ceftriaxone, cefotaxime), carbapenem (meropenem) monobactam
Cytoplasmic membrane	**Bactericidal agents**, considered at the time to be the last line of antibiotics in infections caused by MDR pathogens due to their unselective site of action, can cause a toxic effect in humans, such as the nephrotoxic or neurotoxic effect caused by polymyxins, limiting their usability only in restricted dosages. Antibiotics enhance cell permeability, resulting in efflux of cellular contents.	**Polymyxins** (for Gram-negatives) **Gramicidin**, **Tyrocidine** (for Gram-positives)
Protein synthesis	Both **bactericidal** and **bacteriostatic** agents target the 50s and the 30s ribosomal subunits, inhibiting protein synthesis.	Acting on 50s: **Macrolides:** Erythromycin **Chloramphenicol** **Oxazolidine** Binding site on 50s/Acting in the 30s: **Tetracycline** **Aminoglycosides:** Streptomycin, Gentamycin, Neomycin, Kanamycin, Tobramycin, Amikacin
Nucleic acid synthesis	**Bactericidal** antibiotics interfering with bacterial DNA and RNA synthesis	Acting on DNA gyrase: **Quinolones:** Nalidixic acid **Fluoroquinolones:** Ciprofloxacin, Norfloxacin, Ofloxacin **Aminocoumarin:** Novobiocin Acting on RNA polymerase: **Rifamycin:** Rifampicin
Folate synthesis	**Bacteriostatic**, inhibit the synthesis of DHF * and THF ^#^	Inhibit PABA ^$^ to DHF *: **Sulfonamide** Inhibit DHF to THF ^#^: **Trimethoprim**

* DHF: dihydrofolate; ^#^ THF: tetrahydrofolate; ^$^ PABA: p-Aminobenzoic acid. Source: https://microbeonline.com/mechanisms-of-action-of-antibiotics-an-overview/ (accessed on 28 October 2024).

**Table 2 antibiotics-14-00222-t002:** Main types of enzymes, reactions, and target antibiotics modified.

Enzyme Family	Enzyme Types	Reaction Catalyzed	Target Molecules
Acetyltransferases (AAC)	AAC, CAT, VAT	Acetylation	Aminoglycosides, Chloramphenicol, Virginiamycin
Phosphotransferase (APH)	APH, CPT	Phosphorylation	Aminoglycosides, Chloramphenicol
Adenylyltransferase (ANT)	ANT, LIN	Adenylylation	Aminoglycosides, Lincosamides
Macrolide Modifying Enzymes	MPH, Glycosyltransferases, Acetyltransferases	Phosphorylation, Glycosylation, Acetylation	Macrolides, Ketolides, Lincosamides, Streptogramins
Fosfomycin Modifying Enzymes	FosA, FosB, FosX, FomA, FomB	Various modifications	Fosfomycin
Rifamycin Modifying Enzymes	ADP-ribosyltransferases, Glycosyltransferases, Phosphotransferases, Monooxygenases	Various modifications	Rifamycins
Tetracycline Modifying Enzymes	TetX, Rox	Hydroxylation, Oxida tion	Tetracyclines
β-lactamases	CTX-M, TEM, SHV, KPC, BLC, OXA, NDM, VIM, IMP	Hydrolysis	β-lactams (Penicillins, Cephalosporins, Carbapenems, Monobactams)
Macrolide Esterases (ME)	EreA, EreB	Hydrolysis	Macrolides

**Table 3 antibiotics-14-00222-t003:** Bacterial strategies for protecting antibiotic target site.

Mechanism	Description	Examples
Decreasing Penetration	Limiting the entry of antibiotics into the cell through complex cell walls and porin mutations.	Reduced porin expression, altered porin function
Efflux Pumps	Active transport proteins expel antibiotics from the cell using energy from ATP hydrolysis or chemical gradients.	ABC transporters, SMR family, MATE family, MFS, RND family

**Table 4 antibiotics-14-00222-t004:** Summary of bacterial strategies for antimicrobial resistance through target site modifications and bypass mechanisms.

Mechanism	Description	Examples
Target Protection	Resistance proteins protect the target site from antibiotics.	TetM, TetO (tetracycline resistance), Qnr (quinolone resistance)
Point Mutation	Mutations in genes encoding the target site reduce antibiotic affinity.	RNA polymerase (RIF resistance), DNA gyrase (FQ resistance)
Enzymatic Alteration	Enzymes modify the target site, reducing antibiotic binding.	Methylation of 23S rRNA (macrolide resistance), Cfr enzyme (linezolid resistance)
Target Replacement	Replacement of normal target proteins with resistant variants.	PBP2a (β-lactam resistance)
Target Bypass	Increased production of target sites or use of alternative pathways.	Overproduction of DHFR or DHPS (TMP-SMX resistance), use of exogenous folate

## Data Availability

Not applicable.

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
