# Peer review of "Acquired Bacterial Resistance to Antibiotics and Resistance Genes: From Past to Future"

_antibiotics, 2025, doi:10.3390/antibiotics14030222_

Round 1

Reviewer 1 Report

Comments and Suggestions for Authors

The manuscript attempts to provide a comprehensive review of the mechanisms of antibiotic resistance. While I appreciate and respect the authors' efforts in tackling such a challenging task, I find the manuscript very difficult to read due to its poor structure and grammar (including word choice, punctuation, and missing principal parts in sentences), as well as confusing or contradictory statements. The principles of antimicrobial activities and mechanisms of antibiotic resistance discussed in the manuscript have already been reviewed and summarized in many other publications. Although the Section 8   (Hypotheses for solutions to antibiotic resistance) does provide a good overview of recent publications.

I provide only a few of many  suggestions:

1.       Abstract: reconsider the wording in L20-23.

2.       The Fig 1  represents a modified diagram published in other sources; the reference to the original diagram must be provided, the same approach must be maintained  for tables and figures throughout of the manuscript.  

3.       Critically assess the information in the Table 1 as it is often confusing. Do “ molecules” represent classes of antibiotics?   What is the intent of the statement “ bactericidal, but toxic to humans”? Are there antibiotics not toxic to humans?

4.       L375; the “the phosphorylation of groups 2'-OH of the molecules 14 and 16 members” is very confusing. Please apply the correct nomenclature in description of chemical compounds.

5.       L380; the statement “2'-OH group of the macrolide ring” is incorrect. The 2'-OH group belongs to the desosamine sugar, not to the macrolide ring.

6.       Throughout the manuscript.  Gene names, as well as the names of bacterial species, must be italicized; the names of bacterial species must be provided in full when used for the first time in the manuscript. The names of ribosomal subunits should always include a capital 'S' (i.e. Svedberg unit).  

7.       L963: please reconsider and reword the sentence.

8.       The manuscript conclusion has statements related to topics not covered in the manuscript (f.e. the therapeutic use of bacteriophages as a tool for the control of bacterial resistance).

Comments on the Quality of English Language

The manuscript will benefit from a scrupulous grammar review. Some sentences are unfinished, or missing principal components.

Author Response

Dear Reviewer,

thank you for giving us the opportunity to submit a revised draft of our manuscript titled: “Acquired bacterial resistance to antibiotics and resistance genes: from past to future”, Manuscript ID: antibiotics-3328094, to Antibiotics.

We appreciate the time and effort that you have dedicated to providing your valuable feedback on our manuscript. We are grateful for your insightful comments on our paper. We have been able to incorporate changes to reflect the suggestions provided by the editor and reviewers. We have highlighted the changes within the manuscript.

Here is a point-by-point response to your comments and concerns.

Comments from Reviewer 1 (R1)

The manuscript attempts to provide a comprehensive review of the mechanisms of antibiotic resistance. While I appreciate and respect the authors' efforts in tackling such a challenging task, I find the manuscript very difficult to read due to its poor structure and grammar (including word choice, punctuation, and missing principal parts in sentences), as well as confusing or contradictory statements. The principles of antimicrobial activities and mechanisms of antibiotic resistance discussed in the manuscript have already been reviewed and summarized in many other publications. Although the Section 8 (Hypotheses for solutions to antibiotic resistance) does provide a good overview of recent publications.

I provide only a few of many suggestions:

R1.1:   Abstract: reconsider the wording in L20-23.

Reply to R1.1 We agree with the referee and have rephrased the sentences according to the kindly suggestions of the reviewer.

R1.2:  The Fig 1 represents a modified diagram published in other sources; the reference to the original diagram must be provided, the same approach must be maintained for tables and figures throughout of the manuscript.  

Reply to R1.2 The requested adjustments have been made: the reference to the original diagram represented in Fig. 1 has been provided, and the same approach has been consistently applied to tables and figures throughout the manuscript. Please let me know if further clarification or verification is needed.

R1.3:   Critically assess the information in the Table 1 as it is often confusing. Do “molecules” represent classes of antibiotics?   What is the intent of the statement “bactericidal, but toxic to humans”? Are there antibiotics not toxic to humans?

Reply to R1.3 According to the referee suggestion, the information in Table 1 could benefit from clarification to avoid potential confusion.

  1. Do “molecules” represent classes of antibiotics?

The term "molecules" might imply individual compounds rather than broad classes. To enhance clarity, it should be specified whether it refers to specific chemical structures (e.g., polymyxins) or overarching antibiotic categories (e.g., peptide antibiotics targeting the cytoplasmic membrane).

  1. What is the intent of the statement “bactericidal, but toxic to humans”?
    This statement likely refers to antibiotics such as polymyxins, which are bactericidal due to their ability to disrupt the cytoplasmic membrane. However, their unselective mechanism of action—affecting both eukaryotic and prokaryotic cytoplasmic membranes—can lead to toxic effects in humans, such as nephrotoxicity and neurotoxicity. This toxicity restricts their use to controlled doses, especially as a last-resort option for infections caused by multidrug-resistant (MDR) pathogens.
  2. Are there antibiotics not toxic to humans?

Most antibiotics have a potential for adverse effects, but their toxicity levels vary. Many antibiotics selectively target bacterial processes (e.g., cell wall synthesis or ribosomal differences) with minimal human toxicity under normal dosages. However, drugs like polymyxins, which lack this selectivity, pose a higher risk of toxicity. To address this point in Table 1, a distinction between highly toxic agents and those with low or manageable toxicity profiles should be included for balance.

By refining the descriptions and specifying these points, Table 1 can be made more precise and easier to interpret.

R1.4:  L375; the “the phosphorylation of groups 2'-OH of the molecules 14 and 16 members” is very confusing. Please apply the correct nomenclature in description of chemical compounds.

Reply to R1.4 We agree with the referee and have rephrased the sentences according to the kindly suggestions of the reviewer.

R1.5:   L380; the statement “2'-OH group of the macrolide ring” is incorrect. The 2'-OH group belongs to the desosamine sugar, not to the macrolide ring.

Reply to R1.5 The statement “2'-OH group of the macrolide ring” is indeed incorrect. The 2'-OH group belongs to the desosamine sugar attached to the macrolide ring, not the macrolide ring itself. Macrolide 2'-phosphotransferases, such as mph(A) and mph(B), act on this 2'-OH group of the desosamine sugar. These enzymes transfer the γ-phosphate of a nucleotide triphosphate to the 2'-OH group, disrupting the macrolide’s interaction with A2058 in the bacterial ribosome and thereby conferring resistance to 14-, 15-, and 16-membered-ring macrolide antibiotics (Fyfe et al., 2016). The text should be revised to reflect this distinction for scientific accuracy.

R1.6: Throughout the manuscript.  Gene names, as well as the names of bacterial species, must be italicized; the names of bacterial species must be provided in full when used for the first time in the manuscript. The names of ribosomal subunits should always include a capital 'S' (i.e. Svedberg unit).  

Reply to R1.6: According to the reviewer's requested changes have been implemented:

Gene names and names of bacterial species have been italicized throughout the manuscript.

Full names of bacterial species have been provided upon their first mention.

The names of ribosomal subunits have been consistently formatted with a capital 'S' (e.g., Svedberg unit).

R1.7:   L963: please reconsider and reword the sentence.

Reply to R1.7 The sentence at L963 has been reworded for improved clarity.

R1.8:   The manuscript conclusion has statements related to topics not covered in the manuscript (f.e. the therapeutic use of bacteriophages as a tool for the control of bacterial resistance).

Reply to R1.8: The conclusion has been revised to ensure that all statements are directly aligned with the topics covered in the manuscript. References to subjects not discussed in the main text, such as the therapeutic use of bacteriophages, have been removed.

R1.9:  The manuscript will benefit from a scrupulous grammar review, Some sentences are unfinished or missing principal components.

Reply to R1.9 We thank the reviewer for the suggestion, the grammar and syntax have been thoroughly reviewed and revised throughout the manuscript to address any unfinished sentences or missing components.

Reviewer 2 Report

Comments and Suggestions for Authors

The review entitled: “Acquired bacterial resistance to antibiotics and resistance genes: from past to future” is one of the day’s hot topics and I think it is worth reviewing the recent research data and discovering to not only understand the current stage of the antimicrobial resistance issue but also pathways to address it. Also, it is important to fine-tune this challenge into the new concept of “One Health” as one of the major components, even though the author has not mentioned it in the whole review.

The author did an excellent job by putting up all these research ideas. However, they only focus on opinions and comments rather than actual research data. The authors gathered a lot of information and sometimes the paragraphs are overcrowded. Nevertheless, some things need to be addressed to improve the quality of this review.

Line 40: there should be a “,” after “these”. …. “among these,  the inhibition of Bacillus anthracis” and please provide reference for this statement.

Line 41 to 44: the sentence is not understandable, “consequently” is creating confusion.

Figure 1: I think the figure title should come after the figure (under the figure)

I can’t see Carbapenem 1st, 2nd and 3rd generation resistance including imipenem. Can the author upgrade this diagram?

2. Resistance or persistence: I think the author should also discuss the concepts of bactericides and bacteriostatic agents along with the persistence and the resistance.

3. Antimicrobial resistance mechanisms:

Line 93-95 not clear. Do you mean dangerously?

Can the author discuss how the overuse of antibiotics directly contributes to the mechanism of resistance, versus the antibiotics' selection of a naturally resistant subpopulation of bacteria?

I agree with the statement in lines 101 to 104, and I don't think the use of antibiotics enables bacteria to develop the mechanism of resistance which existed way back before the discovery of antibiotics.

From line 155: the author can introduce the concept of horizontal versus vertical transfer of resistance genes.

Line 167-167: I think this is the predominant way of resistance occurrence.

Again figure 2: the title should come after the figure.

The section i)Structural modifications of the antimicrobial molecule: lack concise examples of genes directly involved. I suggest that the author include examples of genes

Ex: 

Line 320 - 322: “There are several modifying enzymes that catalyze the reactions through acetylation (aminoglycosides, chloramphenicol, streptogramins), phosphorylation (aminoglycosides, chloramphenicol), and adenylation (aminoglycosides, lincosamides)” 

These are not enzymes but drug families otherwise, the sentence is unclear.

Line 341-344: what about the geographical location?

Line 366, remove one comma

Line 380: while streptogramin

Overall section i) is too confusing and should be separated by group of ATB or a table is needed to organize the whole idea.

I suggest that the author entitle some of the paragraphs, especially those starting with “Another example….” , and “a further example ..”  it makes it easier to follow the introduction of a new idea.

Line 649: Do you mean the host (the environment)? Confusing

Line 698: tetracycline belongs to the cycline class of antibiotics, including Doxycycline, Eravacycline etc ..

Line 699: not fluoroquinolone but quinolone class, 

Line 700: Vancomycin belongs to glycopeptides class, colistin (peptide) etc …

Line 712: sulfonamide or sulphonamide??

Genes should be in italic font

Line 726: bacteria name should be in italic

Line 728 - 733: both mechanisms can now be found in  gram neg and gram pos: the statement should be revised

Again, I think there is a need for a table summarizing all

Line 825: to,  not toto

Line 840: the author is reintroducing the biofilm formation here. Too much information kills the information.

I strongly recommend that the authors reorganize the paragraphs and subsections and include tables to summarize some concepts, such as resistance genes and ATB classes.

Comments on the Quality of English Language

I am not an English native, but I think there some typo that need to be correct

Author Response

Dear Reviewer,

thank you for giving us the opportunity to submit a revised draft of our manuscript titled: “Acquired bacterial resistance to antibiotics and resistance genes: from past to future”, Manuscript ID: antibiotics-3328094, to Antibiotics.

We appreciate the time and effort that you have dedicated to providing your valuable feedback on our manuscript. We are grateful for your insightful comments on our paper. We have been able to incorporate changes to reflect the suggestions provided by the editor and reviewers. We have highlighted the changes within the manuscript.

Here is a point-by-point response to your comments and concerns.

Comments from Reviewer 2 (R2)

The review entitled: “Acquired bacterial resistance to antibiotics and resistance genes: from past to future” is one of the day’s hot topics and I think it is worth reviewing the recent research data and discovering to not only understand the current stage of the antimicrobial resistance issue but also pathways to address it. Also, it is important to fine-tune this challenge into the new concept of “One Health” as one of the major components, even though the author has not mentioned it in the whole review.

The author did an excellent job by putting up all these research ideas. However, they only focus on opinions and comments rather than actual research data. The authors gathered a lot of information and sometimes the paragraphs are overcrowded. Nevertheless, some things need to be addressed to improve the quality of this review.

R2.1: Line 40: there should be a “,” after “these”. …. “among these, the inhibition of Bacillus anthracis” and please provide reference for this statement.

Reply to R2.1: We agree with the referee and have added references as requested.

R2.2: Line 41 to 44: the sentence is not understandable, “consequently” is creating confusion.

Reply to R2.2: We agree with the referee and have rephrased the sentences according to the kindly suggestions of the reviewer.

R2.3: Figure 1: I think the figure title should come after the figure (under the figure)

I can’t see Carbapenem 1st, 2nd and 3rd generation resistance including imipenem. Can the author upgrade this diagram?

Reply to R2.3: We sincerely thank the reviewer for their insightful comment. In response, we have incorporated carbapenems into the diagram, though not all generations are represented. We trust the reviewer will understand that this diagram is intended to provide focused information without being overly detailed, similar to other published diagrams of this nature. We have made modifications to the diagram, carefully balancing space constraints and integrating feedback from other reviewers.

R2.4: Resistance or persistence: I think the author should also discuss the concepts of bactericides and bacteriostatic agents along with the persistence and the resistance.

Reply to R2.4:  We agree with the referee and have rephrased the sentences according to the kindly suggestions of the reviewer, discussing the concept as requested.

R2.5: Antimicrobial resistance mechanisms:

Line 93-95 not clear. Do you mean dangerously?

Can the author discuss how the overuse of antibiotics directly contributes to the mechanism of resistance, versus the antibiotics' selection of a naturally resistant subpopulation of bacteria?

Reply to R2.5: We acknowledge the reviewer’s comment and appreciate the opportunity to clarify. We refer to the danger posed by the development of bacterial resistance mechanisms following the clinical use of new antibiotics. This concern was a key factor leading the World Health Organization (WHO) to develop the Global Action Plan on Antimicrobial Resistance in 2015.

R2.6: I agree with the statement in lines 101 to 104, and I don't think the use of antibiotics enables bacteria to develop the mechanism of resistance which existed way back before the discovery of antibiotics.

Reply to R2.6: We thank the reviewer for the comment. In the literature there are studies conducted by Perry et al., 2016 (doi: 10.1101/cshperspect.a025197) and D'Costa et al., 2011(doi: 10.1038/nature10388) supporting the theory that bacteria already possessed resistance mechanisms before the discovery and introduction of antibiotics into clinical practice, indicating antibiotic resistance is intrinsic, ancestral and deeply rooted in the microbial pangenome, and a recent phenomenon. In addition, several studies have argued that increasing antibiotic concentrations, even at sub-inhibitory or sub-lethal concentrations, can result in an increase in antibiotic resistant bacteria (ARBs), and antibiotic resistance genes (ARGs) in exposed bacterial communities (Shen et al., 2019 doi: 10.1016/j.envint.2019.105031; Berglund, 2015 doi:10.3402/iee.v5.28564; Zhu et al., 2013 doi: 10.1073/pnas.1222743110). It is therefore believed to give value to studies that argue that the ancestral development of resistance genes began millions of years earlier in environments populated by a myriad of different microorganisms engaged in constant relationships and competition, influenced by the presence of natural antibiotics, and that over time they evolved and selected according to the environmental pressures to which they were exposed (including the use of synthetic antibiotics) (Agudo e Reche, 2024 doi: 10.3389/fmicb.2024.1445155) Indeed, Sandegren and Andersson, 2009 (doi: 10.1038/nrmicro2174) state that antibiotic resistance may have arisen through gene duplication and genetic diversification through mutations over time leading to the emergence of proteins with distinct physiological functions and their expression may be inducible by selective pressure. Thus, precursor proteins possess an affinity for specific antibiotics, gradually evolving into a robust antibiotic-protein interaction and eventually giving rise to resistance mechanisms, resulting in the efficient resistance genes observed in resistomes today (Dantas and Sommer, 2012 doi: 10.1016/j.mib.2012.07.004 ).

R2.7: From line 155: the author can introduce the concept of horizontal versus vertical transfer of resistance genes.

Reply to R2.7: We agree with the referee and have rephrased the sentences according to the kindly suggestions of the reviewer.

R2.8: Line 167-167: I think this is the predominant way of resistance occurrence.

Reply to R2.8: We agree with the referee and have rephrased the sentences according to the kindly suggestions of the reviewer. “The indiscriminate use of antibiotics and/or the use of low concentrations of anti-crocodiles (sub-inhibitors) provides an important contribution to the selection of hypermutable strains (increased mutation rate) and increase the incidence of acquired resistance by bacteria to antimicrobial agents”

R2.9: Again figure 2: the title should come after the figure.

Reply to R2.9: We agree with the referee and have rephrased the sentences according to the kindly suggestions of the reviewer

R2.10: The section i)Structural modifications of the antimicrobial molecule: lack concise examples of genes directly involved. I suggest that the author include examples of genes

Ex: 

Line 320 - 322: “There are several modifying enzymes that catalyze the reactions through acetylation (aminoglycosides, chloramphenicol, streptogramins), phosphorylation (aminoglycosides, chloramphenicol), and adenylation (aminoglycosides, lincosamides)” 

These are not enzymes but drug families otherwise, the sentence is unclear.

Reply to R2.10: We agree with the referee and have rephrased the sentences according to the kindly suggestions of the reviewer, naming the enzymes responsible for catalyzing the reaction for each antibiotic mentioned

R2.11: Line 341-344: what about the geographical location?

Reply to R2.11: We agree with the referee and have rephrased the sentences according to the kindly suggestions of the reviewer, adding the required information

R2.12: Line 366, remove one comma

Reply to R2.12: We agree with the referee and have corrected the manuscript as requested.

R2.13: Line 380: while streptogramin

Reply to R2.13: We agree with the referee and have corrected the manuscript as requested.

R2.14: Overall section i) is too confusing and should be separated by group of ATB or a table is needed to organize the whole idea.

I suggest that the author entitle some of the paragraphs, especially those starting with “Another example….” , and “a further example ..”  it makes it easier to follow the introduction of a new idea.

Reply to R2.14: According to the kind suggestion of the reviewer, we have reorganized the section into smaller, titled paragraphs and included tables to better summarize the information presented in Section I.

R2.15: Line 649: Do you mean the host (the environment)? Confusing

Reply to R2.15: We agree with the referee and have rephrased the sentences to clarify the mean.

R2.16: Line 698: tetracycline belongs to the cycline class of antibiotics, including Doxycycline, Eravacycline etc .

Reply to R2.16: We agree with the referee and have corrected the manuscript as requested.

R2.17: Line 699: not fluoroquinolone but quinolone class, 

Reply to R2.17: We agree with the referee and have corrected the manuscript as requested.

R2.18: Line 700: Vancomycin belongs to glycopeptides class, colistin (peptide) etc …

Reply to R2.18: We agree with the referee and have corrected the manuscript as requested.

R2.19: Line 712: sulfonamide or sulphonamide??

Reply to R2.19: We agree with the referee and have corrected the manuscript as requested, it’s sulfonamide

R2.20: Genes should be in italic font

Reply to R2.20: We agree with the referee and have corrected the manuscript as requested.

R2.21: Line 726: bacteria name should be in italic

Reply to R2.21: We agree with the referee and have corrected the manuscript as requested.

R2.22: Line 728 - 733: both mechanisms can now be found in gram neg and gram pos: the statement should be revised

Reply to R2.22: We agree with the referee and have revised and modified the sentences of the sentences of paragraph

R2.23: Again, I think there is a need for a table summarizing all

Reply to R2.: We agree with the referee and have added a table summarizing section III.

R2.24: Line 825: to, not toto

Reply to R2.24: We agree with the referee and have corrected the manuscript as requested.

R2.25: Line 840: the author is reintroducing the biofilm formation here. Too much information kills the information.

Reply to R2.25: We agree with the referee and have rephrased the sentences according to the kindly suggestions of the reviewer.

 R2.26: I strongly recommend that the authors reorganize the paragraphs and subsections and include tables to summarize some concepts, such as resistance genes and ATB classes.

Reply to R2.26: We agree with the referee and have summarize the information proposed in tables.

Reviewer 3 Report

Comments and Suggestions for Authors

The Authors aimed to discuss data regarding the acquired bacterial resistance to antibiotics and resistance genes from past to future. With minor additions in the manuscript, the paper could be considered for publication:

1. Add one paragraph for testing procedures for various antimicrobials and their contribution scale for antimicrobial resistance.

2. Introduction can be enriched with the latest resistance pattern and their combination with different resistance mechanisms needed to be added in the draft by adding content and references from given articles.doi.org/10.1128/cmr.00064-16, doi.org/10.3389/fmed.2021.677720, etc.

3. Add more data related to one health approach for antimicrobial resistance

4. The entire manuscript requires more language and grammar correction

Comments on the Quality of English Language

The entire manuscript requires more language and grammar correction

Author Response

Dear Reviewer,

thank you for giving us the opportunity to submit a revised draft of our manuscript titled: “Acquired bacterial resistance to antibiotics and resistance genes: from past to future”, Manuscript ID: antibiotics-3328094, to Antibiotics.

We appreciate the time and effort that you have dedicated to providing your valuable feedback on our manuscript. We are grateful for your insightful comments on our paper. We have been able to incorporate changes to reflect the suggestions provided by the editor and reviewers. We have highlighted the changes within the manuscript.

Here is a point-by-point response to your comments and concerns.

Comments from Reviewer 3 (R3)

The Authors aimed to discuss data regarding the acquired bacterial resistance to antibiotics and resistance genes from past to future. With minor additions in the manuscript, the paper could be considered for publication:

R3.1: Add one paragraph for testing procedures for various antimicrobials and their contribution scale for antimicrobial resistance.

Reply to R3.1: We agree with the referee and have added one paragraph for testing procedures for various antimicrobials and their contribution scale for antimicrobial resistance according to the kindly suggestions of the reviewer.

R3.2: Introduction can be enriched with the latest resistance pattern and their combination with different resistance mechanisms needed to be added in the draft by adding content and references from given articles.doi.org/10.1128/cmr.00064-16, doi.org/10.3389/fmed.2021.677720, etc.

Reply to R3.2: We agree with the referee and have rephrased the introduction according to the kindly suggestions of the reviewer.

R3.3: Add more data related to one health approach for antimicrobial resistance

Reply to R3.3: We agree with the referee and have added more data related to one health approach for antimicrobial resistance according to the kindly suggestions of the reviewer.

R3.4: The entire manuscript requires more language and grammar correction

Reply to R3.4: Language editing has been carefully conducted thoroughly.

Reviewer 4 Report

Comments and Suggestions for Authors

The article is a very good review of antimicrobial resistance. However, the introduction does not provide enough background and justification for this review. The intro needs to be buttressed. Generally, the article has to be thoroughly edited to improve readability.

Line 41: Please correct the spelling of Penicillium.

Figure 1: Please correct “Saversan” and “Sarvesan” to read “Salvarsan”

Figure 1: Also correct the spelling of penicillin.

Table 1: For protein synthesis, I do not think Oxazolidine acts on 30S subunit of the ribosome. Please clarify this. The 50S should fit here.

Lines 88-90: Could you provide some official statistics for the impacts, such as the number of infections and costs? “The emergence of antimicrobial resistance has greatly affected the impact of infectious diseases, in terms of the number of infections, as well as increased healthcare costs”

Line 109: Please correct “totelluric” to read “to telluric”

Line 222: Authors should provide brief descriptions differentiating the classes of integron.

Line 307: Consider correcting the title of Figure 2 to “Graphical representation of the major mechanisms of antimicrobial resistance”

Lines 358-359: The sentence is not complete. “The genes AAC…”

Lines 424-434: A Table will be more appropriate to help discuss these classes.

Lines 436-438: Please complete this sentence.

Line 718: Correct “Furthermore”

Comments on the Quality of English Language

The quality of English should be improved

Author Response

Dear Reviewer,

thank you for giving us the opportunity to submit a revised draft of our manuscript titled: “Acquired bacterial resistance to antibiotics and resistance genes: from past to future”, Manuscript ID: antibiotics-3328094, to Antibiotics.

We appreciate the time and effort that you have dedicated to providing your valuable feedback on our manuscript. We are grateful for your insightful comments on our paper. We have been able to incorporate changes to reflect the suggestions provided by the editor and reviewers. We have highlighted the changes within the manuscript.

Here is a point-by-point response to your comments and concerns.

Comments from Reviewer 4 (R4)

R4.1: The article is a very good review of antimicrobial resistance. However, the introduction does not provide enough background and justification for this review. The intro needs to be buttressed. Generally, the article has to be thoroughly edited to improve readability.

Reply to R4.1: We sincerely thank the reviewer for the thoughtful comment and valuable suggestions. However, the authors have opted to keep the introduction concise to prevent the manuscript from becoming overly dense. As this is a review, our intention was to provide a brief introductory overview in the introduction and address the topic comprehensively in the main text.

R4.2: Line 41: Please correct the spelling of Penicillium.

Reply to R4.2: We agree with the referee and have corrected the manuscript as requested.

R4.3: Figure 1: Please correct “Saversan” and “Sarvesan” to read “Salvarsan”

Reply to R4.3: We agree with the referee and have corrected the Figure 1 as requested.

R4.4: Figure 1: Also correct the spelling of penicillin.

Reply to R4.4: We agree with the referee and have corrected the Figure 1 as requested.

R4.5: Table 1: For protein synthesis, I do not think Oxazolidine acts on 30S subunit of the ribosome. Please clarify this. The 50S should fit here.

Reply to R4.5: We thank the reviewer for the opportunity to clarify this important and interesting point.The mode of action of oxazolidinones is not completely clear, however, it has been it has been shown that the binding site of the oxazolidinones is on 50S, although they don't inhibit peptidyl transferase like chloramphenicol and lincomycin (Lin et al., 1997 https://doi.org/10.1128/aac.41.10.2127; Zhou et al., 2002 https://doi.org/10.1128/aac.46.3.625-629.2002). In line with some studies in the literature it is considered palausible that, although the binding site of the oxazolidinones is on 50S, they inhibit formation of the initiation complex, which is composed of 30S subunit, fMet-tRNA, mRNA, GTP and initiation factors IF1–3, therefore the "functional site of action" of these antibiotics is located on the 30S subunit (Okuyama et al., 1971 https://doi.org/10.1016/S0006-291X(71)80106-7; Swaney et al., 1998 https://doi.org/10.1128/aac.42.12.3251). According to Bobkova et al. 2003 (doi 10.1074/jbc. M209249200) oxazolidinones bind between the P and A loops, partially overlapping the P site of the peptidyltransferase, thus inhibiting the binding of the initiator tRNA to the P site of the peptidyltransferase on 50 S and the formation of the first peptide bond.

Thus we can conclude that oxazolidinones have action both on the 30s unit where they inhibit the formation of the initiation complex, and on the 50s unit where they interfere with the binding of the initiator fMet-tRNA to the P site of the ribosomal peptidyltransferase center.

However, in order to make the information more complete, if the reviewer agrees, we can report both subunits, as described in the literature.

R4.6: Lines 88-90: Could you provide some official statistics for the impacts, such as the number of infections and costs? “The emergence of antimicrobial resistance has greatly affected the impact of infectious diseases, in terms of the number of infections, as well as increased healthcare costs”

Reply to R4.6 We agree with the referee and have provided the requested data according to the kindly suggestions of the reviewer.

R4.7: Line 109: Please correct “totelluric” to read “to telluric”

Reply to R4.7: We agree with the referee and have corrected the manuscript as requested.

R4.8: Line 222: Authors should provide brief descriptions differentiating the classes of integron.

Reply to R4.8 We agree with the referee and have provided the requested data according to the kindly suggestions of the reviewer.

R4.9: Line 307: Consider correcting the title of Figure 2 to “Graphical representation of the major mechanisms of antimicrobial resistance”

Reply to R4.9: We agree with the referee and have changed the title of Figure 2.

R4.10: Lines 358-359: The sentence is not complete. “The genes AAC…”

Reply to R4.10 We agree with the referee and have corrected

R4.11: Lines 424-434: A Table will be more appropriate to help discuss these classes.

Reply to R4.11: We agree with the referee and have added a table to summarize these classes as suggested.

R4.12: Lines 436-438: Please complete this sentence.

Reply to R4.12: We agree with the referee and have completed the sentence

R4.13: Line 718: Correct “Furthermore”

Reply to R4.13: We agree with the referee and have corrected

Reviewer 5 Report

Comments and Suggestions for Authors

Dear Authors,

I read your paper with great interest and I find it relevant to the current pressing issue of antimicrobial resistance spreading all over the world.

In fact, I did not find any major, or even minor issues in the paper. It provides comprehensive information on AMR, accordingly with the title - from the past and ends with the future prospects.

Therefore my recommendation is to accept the manuscript in its present form

Author Response

Dear Reviewer,

thank you for giving us the opportunity to submit a revised draft of our manuscript titled: “Acquired bacterial resistance to antibiotics and resistance genes: from past to future”, Manuscript ID: antibiotics-3328094, to Antibiotics.

We appreciate the time and effort that you have dedicated to providing your valuable feedback on our manuscript. We are grateful for your insightful comments on our paper. We have been able to incorporate changes to reflect the suggestions provided by the editor and reviewers. We have highlighted the changes within the manuscript.

Here is a point-by-point response to your comments and concerns.

Comments from Reviewer 5 (R5)

Dear Authors,

I read your paper with great interest and I find it relevant to the current pressing issue of antimicrobial resistance spreading all over the world.

In fact, I did not find any major, or even minor issues in the paper. It provides comprehensive information on AMR, accordingly with the title - from the past and ends with the future prospects.

Therefore my recommendation is to accept the manuscript in its present form

Reply to R5: Thank you for your thorough and considerate evaluation of our manuscript. We are pleased to hear that you found our work relevant to the critical issue of antimicrobial resistance (AMR) and that it aligns with your expectations regarding the title and content. Your positive feedback and recommendation to accept the manuscript in its current form are greatly appreciated. It is gratifying to know that you found the information presented to be both comprehensive and insightful, spanning historical perspectives and future prospects of AMR. Thank you once again for your kind and supportive review.

Round 2

Reviewer 1 Report

Comments and Suggestions for Authors

Thank you for the opportunity to review you revised manuscript. Unfortunately, I still find it difficult to follow due to its somewhat illogical structure and grammar. Additionally, the manuscript reiterates content from already published reviews and does not contribute new knowledge to the field.

Comments on the Quality of English Language

Needs improvement.